# A stochastic daily weather generator for perennial crop simulations in tropical Malaysia

Christopher Boon Sung Teh[1,2]*, See Siang Cheah[3], David Ross Appleton[4]

**1** Faculty of Agriculture, Universiti Putra Malaysia, Serdang, Malaysia, **2** Institute of Plantation Studies, Universiti Putra Malaysia, Serdang, Malaysia, **3** Research & Development, Dami Oil Palm Research Station, New Britain Palm Oil Limited, Mosa, West New Britain Province, Papua New Guinea, **4** SD Guthrie Research Sdn Bhd, Chemara Research Centre, Lot 2664, Jalan Pulau Carey, Carey Island, Kuala Langat, Selangor, Malaysia

* chris@upm.edu.my

## Abstract

Weather generators are crucial for agricultural modeling in tropical regions, where historical weather data are often scarce or incomplete. This study introduces Msia-Gen, a stochastic daily weather generator for Malaysia's tropical climate, emphasizing computational simplicity, site-specific parameterization, and p ractical applicability. The model was calibrated using data from 12 sites across Malaysia and validated at 11 independent sites, encompassing diverse climatic conditions from Peninsular to East Malaysia. MsiaGen uses a Skew Normal distribution for air temperatures to capture observed asymmetries, particularly in maximum temperatures, while utilizing Weibull and Gamma distributions for wind speed and rainfall, respectively. The generator incorporates first-order autoregressive processes for temporal dependencies and a two-state Markov chain for wet/dry day sequencing. Validation showed strong monthly-scale performance, with mean absolute errors below 1.2% for temperatures, 2.4% for wind speed, and 1.8% for rainfall, along with near-zero model bias and high overall model agreement scores (Kling-Gupta Efficiency metric >0.8). Daily scale validation using quantile-quantile plots revealed excellent agreement for temperature distributions, with points clustering tightly along the identity line within common ranges (21–28 °C for minimum and 25–39 °C for maximum temperatures). Empirical cumulative distribution function analysis indicated that $85 \pm 10\%$ of daily temperature errors were within ±2.0°C, $94 \pm 6\%$ of wind speed errors were within ±1.0 m s$^{-1}$, and $83 \pm 5\%$ of rainfall errors were within ±20 mm. However, performance declined for extreme events, particularly rainfall exceeding 80–100 mm and wind speeds above 3–4 m s$^{-1}$, likely due to distribution tail limitations and short observational records (3–5 years). Further validation using oil palm yield simulations at two independent plantation sites demonstrated that generated weather reproduced temporal dynamics across multiple planting densities. MsiaGen offers a practical and data-efficient tool for tropical agricultural research.

**Data availability statement:** Full source code of model will be made available for public access at: https://github.com/cbsteh/MsiaGen.jl.

**Funding:** This study is funded by Universiti Putra Malaysia under the grant "Research Incentive Grant for Teaching and Learning (GIPP) 2024", Project code no. 9323794.

**Competing interests:** The authors have declared that no competing interests exist.

## Introduction

Weather generators (WGs) are vital tools in agricultural modeling. They provide synthetic daily weather data crucial for long-term crop productivity assessments, climate change impact studies, and agricultural risk analyses [1,2]. Relying on a single historical weather sequence, regardless of its duration, captures only one instance of climate variability, whereas crop models for yield prediction, risk evaluation, and management optimization demand multiple scenarios to account for diverse climatic outcomes. WGs fulfill this by generating statistically consistent realizations, facilitating uncertainty quantification in crop yields, probabilistic extreme event assessments, management strategy optimization, and climate adaptation evaluations [3–7].

Many established WGs, such as USCLIMATE [8], WXGEN [9], LARS-WG [10], CLIMGEN [11], and CLIGEN [12], have been predominantly developed for temperate climates. Recent efforts have applied advanced WGs in Malaysia, a tropical country predominantly characterized by a Köppen Af (tropical rainforest) climate [13]. The Advanced Weather Generator (AWE-GEN) [14], for instance, was tested on several Malaysian locations [15–17]. By incorporating the Neyman-Scott Rectangular Pulse model, AWE-GEN demonstrated proficiency in simulating extreme rainfall events using hourly data, making it valuable for flood risk assessment. Yet, the reliance on hourly weather data by AWE-GEN presents a prohibitive barrier to its application in Malaysian agriculture in general. Daily weather records, let alone hourly records, are not always available because weather stations in Malaysia are predominantly located in urban areas, leaving many agricultural regions without nearby observations [18–21]. Rural areas often lack weather records, or even if they exist, they are frequently incomplete.

What is required for Malaysian agricultural applications is a WG that addresses specific practical requirements: (i) operation using only daily weather data, (ii) preservation of site-specific characteristics, (iii) computational simplicity, and (iv) easily modifiable parameters for scenario analysis. The validation of such WGs for tropical agricultural applications is particularly valuable for perennial crops. Oil palm (*Elaeis guineensis* Jacq.) is Malaysia's most economically vital crop, contributing 30% of global palm oil production and generating over RM95 billion (USD22 billion) in export revenue annually [22]. The crop's physiology exhibits distinct responses to meteorological conditions, making it a suitable test case for WG validation. Oil palm maintains optimal photosynthetic activity within a narrow temperature range (26–28 °C), with productivity declining sharply above 35 °C [23]. Its prolonged fruit development cycle (5–6 months) establishes lagged relationships, where past weather conditions affect current yields [24]. Furthermore, the shallow root system of the crop renders it particularly sensitive to rainfall distribution patterns rather than total accumulation, leading to typical yield fluctuations of 20–30% between years due to climatic factors [25].

Consequently, to address this need, this study aimed to 1) develop a daily stochastic WG, called *MsiaGen*, specifically for Malaysia's tropical climate, 2) validate its accuracy using a suite of statistical metrics and visual diagnostics, and 3) evaluate its performance in oil palm yield simulations by comparing outcomes driven

by generated versus observed weather data. MsiaGen generates synthetic weather by either extracting the necessary statistical properties from observed daily weather data or allowing users to directly specify these properties to define the desired probability distribution shapes. MsiaGen's design philosophy emphasizes accessibility, computational efficiency for the rapid generation of daily weather, and a transparent framework for parameter modification. Its ability to work with more commonly available daily data makes it suitable for agricultural researchers and practitioners who favor site-specific analysis over complex methods for interpolating weather data between distant stations or other data-intensive approaches.

## Materials and methods

### Study site

Twenty-three meteorological stations (sites) across Malaysia were selected: 16 sites in Peninsular Malaysia and seven sites in Sabah and Sarawak in East Malaysia (Fig 1 and Table 1). Site selection prioritized stations that spanned across the country, but crucially, that these sites had complete daily records of minimum/maximum air temperatures, wind speed, and rainfall. However, to prevent bias from uneven data lengths, all records were standardized to a 3–5 year period (1095–1827 days). This ensured balanced representation across regions, avoiding disproportionate influence from stations with longer historical data. Raw weather data were obtained from SD Guthrie Berhad (proprietary; accessed under a data-sharing agreement) and from the Malaysian Meteorological Department via the MyMET Data Portal (mymetdata.met.gov.my; paid access with registration). The sites were divided into calibration (n = 12) and validation (n = 11) datasets. The calibration dataset was used for model development and parameterization, while the validation dataset was used for unbiased accuracy testing.

Malaysia is predominantly characterized by Köppen Af (tropical rainforest) climate [13], although Am (tropical monsoon) climate zones can be found as small enclaves, mainly along the northwestern coast of Peninsular Malaysia, including locations such as Alor Setar (Kedah), parts of Penang, and Perlis [26,27]. State-wide reference evapotranspiration (ET0) ranges from approximately 1200–1700 mm year$^{-1}$, and the climatic water balance is generally positive (approximately 200–2500 mm year$^{-1}$), with substantial state-to-state variation. ET0 values are from the Global Aridity Index and Potential Evapotranspiration Climate Database v3.1 (FAO-56 Penman-Monteith; 1970–2000 monthly climatology, 30 arc-sec), and the climatic water balance was computed as precipitation minus ET0 using the WorldClim v2 monthly precipitation climatology (1970–2000) [28].

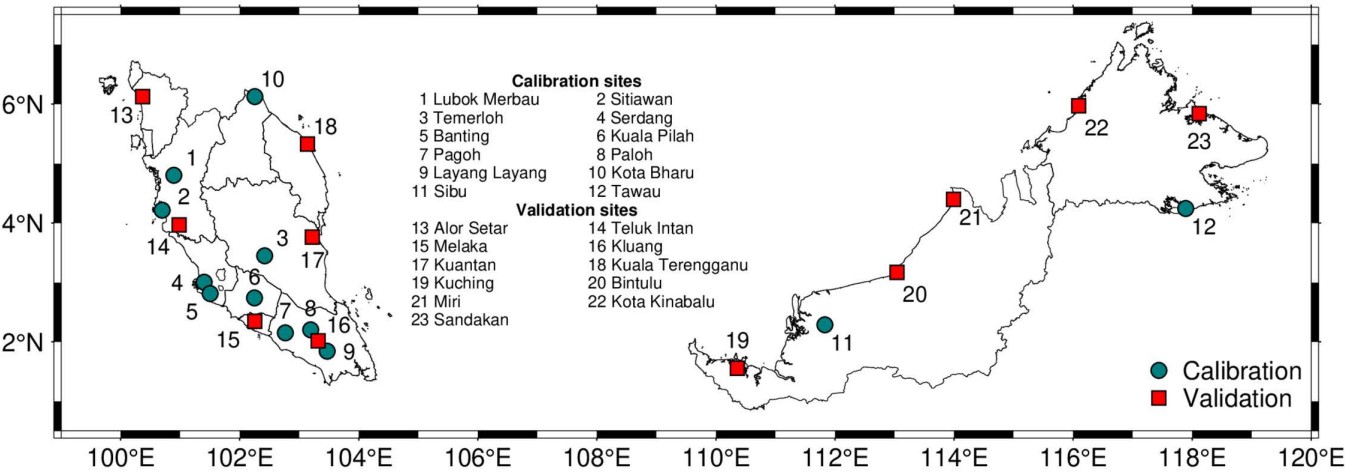

**Fig 1. A total of 23 meteorological stations across Malaysia were selected for this study, comprising 16 sites in Peninsular Malaysia and seven in East Malaysia.** Of these, 12 and 11 stations were used for the model calibration and validation, respectively.

**Table 1. Mean annual (±1 SD) minimum (Tmin) and maximum (Tmax) air temperatures, wind speed, and total rainfall at each site. The number of years indicates the period of record, and total days (N) and wet days (Nwet) are counts across all years for each site. The sites were grouped for model calibration and validation.**

| Site (°N, ° E) | Years | days N | °C Tmin | °C Tmax | m s$^{-1}$ Wind | mm Rainfall | days $N_{wet}$ |
|---|---|---|---|---|---|---|---|
| *Calibration sites* | | | | | | | |
| Lubok Merbau (4.8046, 100.8917) | 2017–19 | 1095 | 23.71±0.87 | 33.19±1.57 | 1.23±0.33 | 1886±331 | 570 |
| Sitiawan (4.2168, 100.6978) | 2017–19 | 1095 | 24.35±0.81 | 32.44±1.53 | 1.23±0.29 | 1937±21 | 516 |
| Temerloh (3.4505, 102.4189) | 2017–19 | 1095 | 23.72±0.87 | 33.43±1.93 | 0.83±0.22 | 2052±769 | 503 |
| Serdang (3.0050, 101.4023) | 2003–07 | 1826 | 23.24±0.67 | 33.07±1.45 | 0.68±0.21 | 2678±372 | 1061 |
| Banting (2.8136, 101.5019) | 2017–19 | 1095 | 23.84±0.82 | 31.98±1.28 | 1.79±0.18 | 1817±278 | 437 |
| Kuala Pilah (2.7389, 102.2480) | 2017–19 | 1095 | 22.89±0.80 | 31.54±1.56 | 1.37±0.42 | 1998±530 | 497 |
| Pagoh (2.1500, 102.7667) | 2017–19 | 1095 | 22.78±0.99 | 31.74±1.68 | 0.75±0.19 | 2304±378 | 543 |
| Paloh (2.4167, 111.2500) | 2017–19 | 1095 | 23.14±0.73 | 32.09±1.84 | 0.95±0.33 | 1169±91 | 560 |
| Layang Layang (1.8426, 103.4692) | 2015–19 | 1826 | 23.74±0.95 | 32.18±1.84 | 1.22±0.64 | 2409±356 | 880 |
| Kota Bharu (6.1248, 102.2544) | 2016–20 | 1827 | 24.51±1.07 | 31.52±1.74 | 2.51±0.82 | 2708±593 | 797 |
| Sibu (2.2873, 111.8305) | 2016–20 | 1827 | 23.75±0.72 | 32.44±1.84 | 2.37±1.13 | 3866±808 | 1182 |
| Tawau (4.2448, 117.8911) | 2015–19 | 1826 | 23.76±0.70 | 31.80±1.19 | 1.72±0.47 | 1861±383 | 843 |
| *Validation sites* | | | | | | | |
| Alor Setar (6.1248, 100.3678) | 2020–24 | 1827 | 24.30±1.04 | 32.73±1.95 | 1.71±0.44 | 2337±106 | 895 |
| Teluk Intan (3.9694, 100.9835) | 2016–20 | 1827 | 24.38±0.80 | 32.57±1.46 | 1.25±0.30 | 1903±193 | 854 |
| Melaka (2.3463, 102.2531) | 2020–24 | 1827 | 24.48±0.88 | 31.95±1.48 | 1.76±0.71 | 2301±298 | 938 |
| Kluang (2.0167, 103.3167) | 2020–24 | 1827 | 24.05±0.78 | 31.87±2.00 | 1.52±0.82 | 2707±188 | 977 |
| Kuantan (3.7634, 103.2202) | 2016–20 | 1827 | 23.99±0.91 | 32.29±1.88 | 2.03±0.78 | 2396±336 | 887 |
| Kuala Terengganu (5.3296, 103.1370) | 2020–24 | 1827 | 24.89±1.00 | 31.49±1.73 | 2.03±0.73 | 3105±702 | 871 |
| Kuching (1.5535, 110.3593) | 2020–24 | 1827 | 23.89±0.76 | 32.09±1.86 | 1.57±0.34 | 4380±414 | 1240 |
| Bintulu (3.1713, 113.0419) | 2016–20 | 1827 | 24.00±0.81 | 31.83±1.54 | 2.21±0.77 | 3768±551 | 1120 |
| Miri (4.3995, 113.9914) | 2016–20 | 1827 | 24.22±0.89 | 31.55±1.27 | 2.69±0.82 | 2791±632 | 930 |
| Kota Kinabalu (5.9714, 116.0952) | 2016–20 | 1827 | 24.45±0.90 | 32.71±1.44 | 2.09±0.52 | 2657±565 | 900 |
| Sandakan (5.8394, 118.1172) | 2020–24 | 1827 | 24.87±0.95 | 31.52±1.70 | 2.17±0.73 | 3077±595 | 975 |

The 23 stations span across Malaysia's tropical climate zones (predominately Köppen Af, with some Am enclaves) with stable air temperatures (mean annual minima 22.9–24.9 °C and maxima 31.5–33.4 °C), low wind speeds (0.7–2.7 m s⁻¹), and high rainfall variability (1169–4380 mm annually, with 146–248 wet days per year) (Table 1) [29]. Sites in East Malaysia exhibited higher wind speeds and rainfall than those in Peninsular Malaysia.

MsiaGen was parameterized independently for each site, capturing local climatology without pooling data or imposing spatial or temporal homogeneity. Because the study's objective was to generate statistically consistent daily weather for agricultural modelling rather than to analyse climatic trends, the framework focused on reproducing contemporary within-month statistics for weather generation. This approach enhances applicability in data-limited settings, where strict homogeneity screening could exclude many stations without improving the quality of short-term stochastic simulations. During calibration, within-month stationarity over the 3–5-year windows was assumed, and formal homogenization tests were not applied because their power is limited for short series [30].

## MsiaGen weather generation framework

MsiaGen generates synthetic daily weather data using a parametric stochastic approach based on probability distributions. The model operates on monthly time steps, requiring statistical parameters (mean, standard deviation, lag-1 autocorrelation, and skewness where applicable) for each calendar month. These parameters can be either extracted from observed daily weather data or directly specified by users to define desired climate scenarios. MsiaGen is written in Julia programming language, and its full source code is available at: github.com/cbsteh/MsiaGen.

**Air temperature generation.** Daily air temperatures exhibit strong temporal persistence, with each day's temperature substantially depending on the previous day's value. To capture this autocorrelation structure, first-order autoregressive AR(1) processes were employed for both minimum (Tmin) and maximum (Tmax) temperatures. The AR(1) model has been extensively validated for temperature simulations across diverse climatic conditions [31,32] and provides an optimal balance between model accuracy and computational simplicity compared with higher-order models [33,34].

For each month $m$, the temperature on day $t$ ($T_t$) is generated as:

$$T_t = c_{T,m} + \rho_{T,m} T_{t-1} + \epsilon_{T,t}^{(m)}$$

(1)

where $\rho_{T,m}$ is the lag-1 autocorrelation coefficient, $c_{T,m}$ is a constant term ensuring the sequence maintains the target mean $\mu_{T,m}$, and $\epsilon_{T,t}^{(m)}$ represents the daily residual term. Analysis of the 12 calibration sites showed that maximum air temperatures often deviated from normality, with two to five months per year having absolute skewness values above 1. Minimum temperatures were generally closer to normal, with only one or two months showing moderate skewness. To accommodate this variability, the residual term $\epsilon_{T,t}^{(m)}$ in Eq. 1 was sampled from a Skew Normal distribution [35], whose parameters (monthly mean, standard deviation, and skewness) were estimated using the method of moments. This distribution provides the flexibility needed for tropical temperature patterns, as it reduces to the standard Normal distribution when the skewness parameter equals zero. The standard deviation of residuals is adjusted for autocorrelation effects to ensure the generated sequence achieves the target variability.

When absolute skewness exceeds 0.995272, the Skew Normal distribution becomes numerically unstable [36]. In such cases, residuals are instead sampled from an F-distribution, which effectively handles extreme right skewness [37]. For left-skewed distributions, values are generated by negating samples from the F-distribution fitted to the absolute skewness value. Detailed equations and parameter derivations are provided in S1 File.

**Wind speed generation.** Wind speed follows a similar AR(1) structure to capture day-to-day persistence. For each month $m$, wind speed on day $t$ ($X_t$) is generated as:

$$X_t = c_{X,m} + \rho_{X,m} X_{t-1} + \epsilon_{X,t}^{(m)}$$

(2)

where the terms parallel those in the temperature model, with $\rho_{X,m}$ representing the lag-1 autocorrelation of wind speed. Wind speed residuals $\epsilon_{X,t}^{(m)}$ are sampled from a Weibull distribution, which has been extensively validated for wind speed modeling applications [38–40]. The Weibull distribution is characterized by shape ($k_m$) and scale ($\lambda_m$) parameters, which are estimated from the monthly mean and standard deviation using empirically derived relationships [41]. Mathematical formulations are provided in S1 File.

**Rainfall generation.** Rainfall generation follows a two-stage process that separately models rainfall amounts and the temporal sequencing of wet (rain ≥0.05 mm) and dry (rain <0.05 mm) days.

**Stage 1: Rainfall amounts.** Daily rainfall amounts for wet days are drawn from a Gamma distribution, a frequent choice in rainfall modeling due to its flexibility in representing positively skewed precipitation data. The distribution is characterized by shape ($s_m$) and scale ($\theta_m$) parameters for each month $m$.

A novel aspect of MsiaGen's approach is the treatment of the Gamma shape parameter as a random variable rather than a fixed monthly value. Analysis of shape parameters fitted to observed wet-day rainfall across all calibration sites, years, and months revealed substantial variability with an asymmetric, right-skewed distribution. To capture this variability, a Generalized Extreme Value (GEV) distribution was fitted to the empirical shape parameter estimates. The GEV distribution is particularly suitable for modeling the heavy-tailed behavior and extremes characteristic of hydrological data.

During weather generation, each wet day's rainfall amount is determined by first sampling a shape parameter from this GEV distribution, then computing the corresponding scale parameter from the target monthly mean wet-day rainfall, and finally drawing the rainfall amount from the resulting Gamma distribution. This approach introduces additional stochastic variability that better represents the natural fluctuation in daily rainfall characteristics. Mathematical details are provided in S1 File.

**Stage 2: Wet and dry day sequencing.** The temporal distribution of wet (W) and dry (D) days follows a first-order two-state Markov chain process, where the probability of rainfall on a given day depends only on the previous day's state. This widely used approach [8–12] is characterized by two transition probabilities: $P_{WW}$ (probability that a wet day follows a wet day, representing persistence of wet conditions) and $P_{WD}$ (probability that a wet day follows a dry day, representing transition from dry to wet conditions).

For each day in the sequence, the model first identifies the previous day's state, then selects the appropriate transition probability ($P_{WW}$ or $P_{WD}$). A random draw from a uniform distribution [0,1] determines the current day's state: if the random value is less than the transition probability, the day is classified as wet; otherwise, it is dry. When a day is determined to be wet, a specific rainfall amount is assigned using the Gamma distribution procedure described above.

**Constraint compliance procedure.** The model employs an iterative approach to ensure that the generated sequences satisfy all user-specified statistical targets, resulting in generated data that closely reflects the observed or required conditions. For each calendar month, the generation algorithms refine their outputs until the statistical properties of the sequences converge to the target values within the set tolerance limits. Air temperature sequences are assessed for mean, standard deviation, lag-1 autocorrelation, and skewness, with iterations continuing until all four metrics deviate by less than 2.5%, or a maximum of 5,000 iterations is reached. Wind speed sequences are evaluated for mean, standard deviation, and lag-1 autocorrelation. Rainfall sequences use a dual-level acceptance process: on a monthly scale, convergence is required within ±2.5% for total rainfall and ±5% for transition probabilities of wet and dry days; on an annual scale, thresholds are set at ±5% for total rainfall and ±10% for transition probabilities. If convergence is not achieved within the iteration limit, the sequence with the smallest maximum deviation from the target statistics is retained.

## Model evaluation

The agreement between model-generated weather and observations was evaluated using various methods, one of which was to use three goodness-of-fit metrics. The first two metrics were the Normalized Mean Absolute Error (NMAE) and Normalized Mean Bias Error (NMBE):

$$NMAE = 100 \times \sum |G_i - O_i| / \sum O_i \qquad (3)$$

$$NMBE = 100 \times \sum (G_i - O_i) / \sum O_i \qquad (4)$$

where $G_i$ and $O_i$ are the model-generated (predicted) and observed (measured) values, respectively, and subscript $i$ denotes the individual data points in the dataset. Both NMAE and NMBE are normalized to remove dependence on the scale of meteorological properties being compared. The third metric employed was the Kling-Gupta Efficiency (KGE) [42], which combines three model agreement properties into a single score: the degree to which model-generated weather ($G$) and observations ($O$) correlate linearly ($r$) with each other and how closely their means ($\mu$) and coefficients of variation ($cv$) match each other. KGE is given by

$$KGE = 1 - \sqrt{(r-1)^2 + (\mu_G/\mu_O - 1)^2 + (cv_G/cv_O - 1)^2} \qquad (5)$$

where KGE scores range between $-\infty$ and +1, with +1 denoting perfect model accuracy.

These three metrics provide complementary assessments of model performance, with each capturing distinct aspects of model agreement. For instance, NMAE measures the average absolute difference between the generated and observed values, whereas NMBE measures the overall model bias. NMAE ranges from 0 to $+\infty$, with NMAE = 0 indicating perfect model accuracy. NMBE ranges from $-\infty$ to $+\infty$, where NMBE = 0 indicates no overall model bias, and increasingly large deviations from zero denote increasing model underestimation (large negatives) or overestimation (large positives). KGE's strengths are two-fold: it is less sensitive to outliers, and it provides a comprehensive evaluation by simultaneously assessing correlation, bias, and variability between generated and observed values.

In addition to the goodness-of-fit metrics, the two-sample Anderson-Darling (AD) test [43] was used as a distributional test to statistically evaluate whether the distributions from the generated and observed data are derived from the same underlying distribution (two-sided; $a = 0.05$). Throughout the paper, the AD rejection rate denotes the proportion of tests with $p < 0.05$.

Model performance was assessed at two temporal scales. At the monthly scale, the three goodness-of-fit metrics, as well as the AD test, compared the mean minimum temperature (Tmin) and maximum temperature (Tmax), mean wind speed, and total rainfall between the model-generated and observed data. This approach focuses on evaluating the model accuracy at the monthly scale by smoothing out daily variability.

Complementing this monthly analysis, the distributional characteristics of daily weather variables were evaluated using visual diagnostic tools. Quantile-quantile (QQ) plots were employed to compare the full distributions of the generated and observed daily data. QQ plots are particularly effective in revealing systematic biases in the model's representation, including those of rare and extreme events, through deviations from the identity (1:1) line. Additionally, plots of the empirical cumulative distribution function (ECDF) were used to determine the frequency and magnitude of daily model errors by showing the proportion of data within specified error thresholds. To further assess the model agreement across the full range of values, hexagonal bin plots (heat maps) were constructed to visually examine the correspondence (agreement) between the generated and observed daily Tmin, Tmax, and wind speed.

**Weather generation for oil palm yield simulations.** Further evaluation of MsiaGen's accuracy and practical utility was conducted using *Sawit.jl*, an oil palm simulation model developed specifically for the Malaysian environment [44]. The evaluation approach involved feeding Sawit.jl with both generated and observed weather data to determine whether the resulting oil palm yield simulations would differ. It is desirable that simulations driven by generated weather match those

driven by observed weather, indicating that the generated weather accurately represents real weather conditions for crop modeling purposes.

Simulations were performed for two oil palm plantation sites in Peninsular Malaysia: Kerayong (3.17 °N, 101.34 °E) and Kalumpong (4.96 °N, 100.57 °E). At Kerayong, the model performance was assessed by comparing 22-year oil palm yield simulations (1993–2014) driven by two distinct weather inputs: observed historical records and MsiaGen's weather-generated output. Fresh fruit bunch (FFB) yield simulations were conducted for three planting densities (136, 160, and 185 palms $ha^{-1}$) and validated against historical measured yields. Kerayong is characterized by a clayey Jawa soil series (Sulfric Endoaquepts) comprising 6.6% sand and 51.4% clay. Climate conditions include a mean annual rainfall of 1824 mm, average annual minimum and maximum temperatures of 23.4 °C and 32.3 °C, respectively, and an average annual wind speed of 1.2 m $s^{-1}$.

Similarly, simulations at Kalumpong were conducted for three planting densities: 124, 138, and 150 palms $ha^{-1}$ over a 23-year period (1997–2019). FFB yield monitoring, however, only started 8–14 years after field planting. Kalumpong features diverse soil types, predominantly Briah (Typic Endoaquepts) and the Jawa soil series, with an average composition of 47.8% sand and 13.7% clay. Kalumpong experiences higher rainfall than Kerayong, with a mean annual precipitation of 2003 mm and similar average annual minimum and maximum temperatures of 23.2 °C and 32.8 °C, respectively, and an average annual wind speed of 1.3 m $s^{-1}$.

Detailed descriptions of the Sawit.jl model, oil palm site characteristics, and field data collection are provided in Teh et al. [44]. Crucially, the historical daily weather data from both Kerayong and Kalumpong sites were excluded from MsiaGen's calibration and validation datasets, ensuring independent model testing. The long-term weather records generated for Kerayong and Kalumpong (22–23 years) provide an additional benefit. They allow for an assessment of the temporal stability of the weather generator model parameters. This is possible because these long-term weather series were generated using parameter values that were originally calibrated on sites with only 3–5 years of data.

**Rainfall sensitivity analysis.** The sensitivity of oil palm yield predictions to changes in rainfall intensity distribution was evaluated using three sets of rainfall manipulation scenarios. Daily rainfall was categorized into lower-intensity (L, ≤ 60 mm $day^{-1}$) and higher-intensity (H, > 60 mm $day^{-1}$) events. The 60-mm threshold corresponds to the 95th percentile of rain-day intensities at both Kerayong and Kalumpong sites. Although H events occurred on only about 5% of rainy days, they contributed 22–25% of annual rainfall at both these sites.

Three sets of graduated manipulations were implemented. In the first scenario set, H events were reduced by 25%, 50%, 75%, and 100%, producing progressively larger rainfall deficits. In the second set, the same graduated reductions of H events were applied, but the removed amounts were redistributed to L events to conserve total annual rainfall. The third set enabled direct comparison of rainfall deficits and redistributions by manipulating an amount equal to 50% of annual H rainfall (approximately 11–13% of annual rainfall). Four scenarios in this third set were considered: (1) removing 50% of H rainfall to impose a deficit; (2) removing an equivalent amount from L rainfall to impose a deficit; (3) reducing H rainfall by 50% with redistribution to L events to conserve rainfall; and (4) removing an equivalent amount from L rainfall with redistribution to H events to conserve rainfall.

All rainfall manipulations were performed using proportional scaling. For reductions, affected events were multiplied by one minus the reduction fraction. When rainfall was redistributed, the removed volume was allocated among recipient events in proportion to each event's relative intensity within that class. This approach preserved the number and timing of rainy days while modifying the intensity distribution, allowing some events to shift between intensity classes (*e.g.*, reducing a 70-mm to 35-mm event moves it from H to L).

The first two sets quantified yield responses to increasing magnitudes of rainfall manipulation, while the third set allowed direct comparison of deficit versus redistribution effects and distinguished the relative importance of higher- and lower-intensity rainfall for oil palm yields.

## Results

### Model performance at monthly temporal scale

MsiaGen demonstrated good performance in reproducing mean monthly air temperatures for both minimum (Tmin) and maximum (Tmax) values (Fig 2a–h). This strong performance was evidenced by consistently high accuracy metrics: absolute errors (NMAE) remained below 1.2% (Fig 2a, e), model bias (NMBE) effectively zero (Fig 2b, f), overall agreement scores (KGE) exceeded 0.8 at most sites (Fig 2c, g), and distributional fidelity was strong, with AD test rejection rates below 2.2% for both temperature variables (Fig 2d, h).

Wind speed generation was similarly robust, with an NMAE below 2.4% (Fig 2i), negligible bias (NMBE effectively zero; Fig 2j), and good overall agreement (KGE above 0.70; Fig 2k). The model did exhibit minor distributional discrepancies at some sites, as reflected by higher AD test rejection rates of 9 and 11% at the validation and calibration sites, respectively (Fig 2l).

Monthly rainfall generation demonstrated strong agreement with observations: NMAE lower than 1.8% (Fig 2m), bias remained within ±2% (Fig 2n), and KGE exceeded 0.95 (Fig 2o). However, the model experienced greater difficulty in replicating the exact distributional characteristics of observed monthly rainfall, as indicated by AD test rejection rates of 16 and 20% for validation and calibration sites, respectively (Fig 2p).

The correlation between mean monthly Tmax and monthly rainfall in the observed data ranged from −0.9 to +0.3 (Fig 3). This relationship reflects the well-established meteorological principle that air temperatures are typically suppressed during periods of increased precipitation due to enhanced cloud cover, reduced solar radiation reaching the surface, and evapotranspiration cooling effects [45]. As the model accurately reproduced both monthly Tmax and rainfall values individually (Fig 2e–h, m–p), the correlation between the generated mean monthly Tmax and monthly rainfall closely matched the observed relationships (Fig 3). This correspondence demonstrates that the model captured individual climatological variables accurately, as well as preserved their underlying physical relationships.

Generated monthly rainfall transition probabilities closely matched observed values at both calibration and validation sites (Fig 4), with low model error and bias (NMAE 6–7%, NMBE 0–4%) and high agreement scores (KGE 0.94–0.96). The model accurately reproduced wet-following-wet probabilities ($P_{WW}$, Fig 4a,c) and dry-following-wet probabilities ($P_{WD}$, Fig 4b,d), with points clustering tightly along the 1:1 line. The validation sites exhibited similar agreement to the calibration sites, indicating stable model performance across different locations.

### Model performance at daily temporal scale

**Quantile-quantile (QQ) and hexagonal bin plots.** QQ plots for daily air temperatures (Tmin and Tmax) reveal a strong agreement between the generated and observed values (Fig 5a, b). Generated quantiles aligned closely with observed values, with points clustering tightly along the identity line for both calibration and validation sites. This alignment was particularly pronounced within the 21–28 °C range for Tmin and 25–39 °C range for Tmax, indicating high fidelity in reproducing these common temperature ranges. Model discrepancies became more apparent at extreme temperatures, with slight departures from the identity line occurring at Tmin values below 21 °C (Fig 5a) and Tmax values below 25 °C (Fig 5b). Despite these minor deviations in the lower tails, the overall representation of the temperature distributions remained highly accurate.

The hexagonal bin plots (heatmaps) further confirmed this strong performance, revealing dense data concentrations (light green, red, and yellow bins) tightly aligned along the identity line for Tmin between 23 °C and 26 °C and Tmax between 30 °C and 35 °C (Fig 6a, b). These optimal performance ranges correspond closely to the most frequently occurring temperature conditions. Although the model demonstrated strong fidelity within these core ranges, individual predictions exhibited varying levels of precision, as indicated by the presence of less dense blue bins. The performance notably declined when the values fell outside these optimal ranges, with increased data scattering particularly evident at temperature extremes.

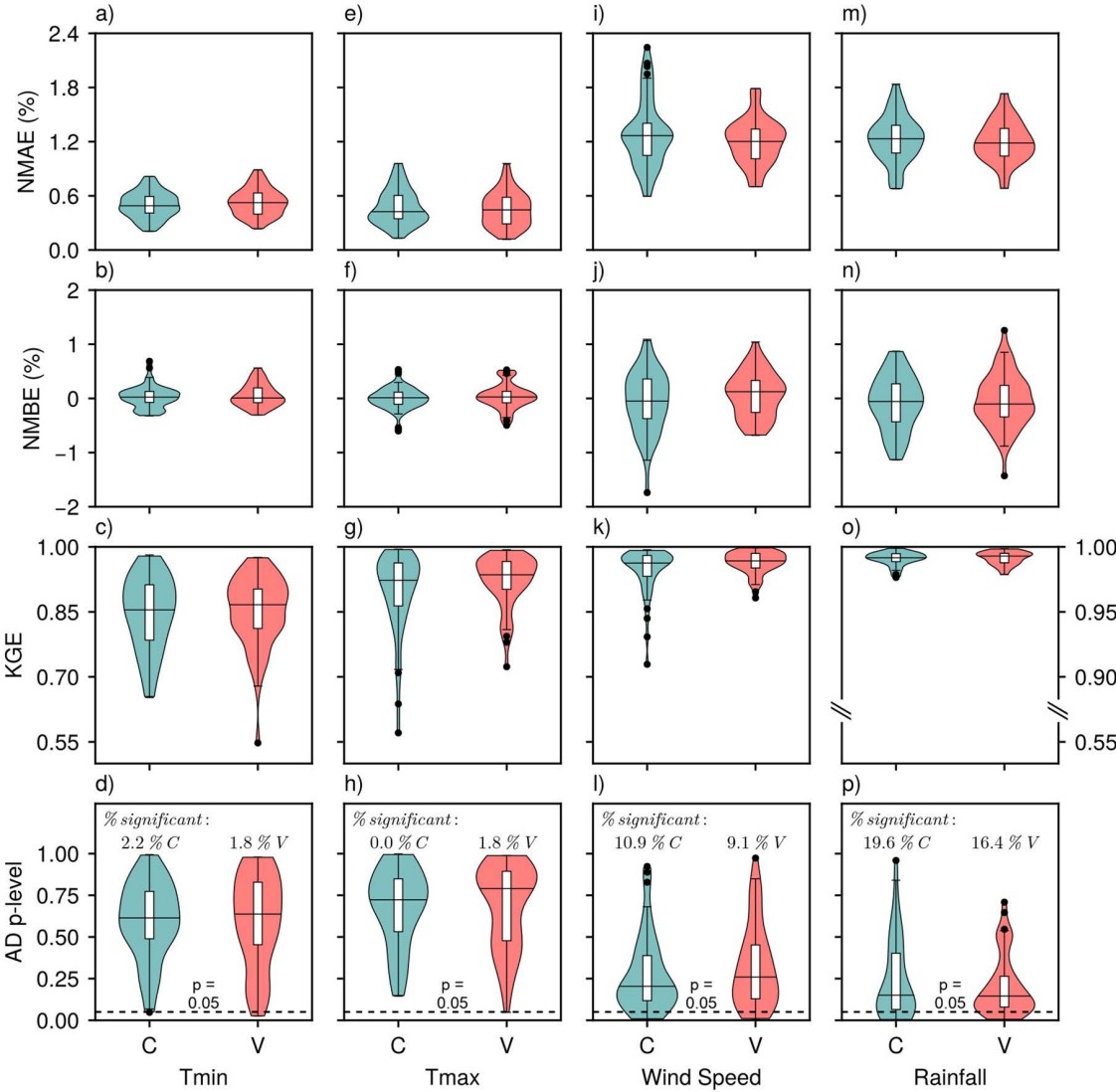

Note: "% significant" indicates the proportion of calibration (C) and validation (V) data that were

significantly different (p<0.05) from observations using Anderson-Darling (AD) test

**Fig 2. Summary of goodness-of-fit metrics (NMAE, NMBE, KGE) and Anderson-Darling test p-values (AD p-level) for mean monthly minimum air temperature (Tmin) (a-d), maximum air temperature (Tmax) (e-h), wind speed (i-l), and total monthly rainfall (m-p), shown separately for the calibration (C) and validation (V) sites (n=46 and 55 site-years, respectively).** NMAE: Normalized Mean Absolute Error; NMBE: Normalized Mean Bias Error; KGE: Kling-Gupta Efficiency.

Daily wind speed generation showed strong agreement for the lower and median quantiles, with generated values clustering tightly along the identity line up to approximately 4 m s$^{-1}$ (Fig 5c). Beyond this threshold, the model underestimated higher wind speed quantiles, indicating a limited capability in reproducing extreme wind events. However, the hexagonal density plot suggests that this threshold was closer to 3 m s$^{-1}$ (Fig 6c), where the densest data concentrations along the identity line occurred, with increased scatter and systematic underestimation becoming apparent at higher wind speeds, particularly >4 m s$^{-1}$.

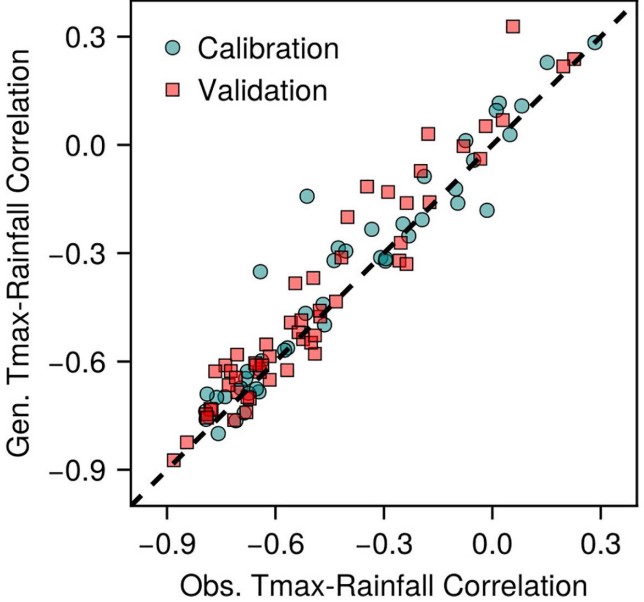

**Fig 3. Comparison of correlations between the mean monthly maximum air temperature (Tmax) and monthly rainfall for the model-generated (gen.) and observed (obs.) data, shown separately for the calibration and validation sites (n = 46 and 55 site-years, respectively).** The dashed line indicates perfect agreement (1:1 line).

Daily rainfall generation captured the distribution well for lower and moderate rainfall amounts (Fig 5d). However, the performance deteriorated at quantiles above 100 mm, where increasingly large deviations from the identity line occurred. The model exhibited a contrasting tendency to overpredict extreme rainfall events at the calibration sites and underpredict them at the validation sites. This divergent behavior likely stems from insufficient sampling of rare extreme events within the relatively short record lengths (3–5 years), compounded by differences in climatological regimes between the site groups. The calibration sites experienced lower average annual rainfall (2267 mm) than the validation sites (2863 mm) (Table 1) and correspondingly lower maximum observed daily rainfall extremes (215 vs. 315 mm). Consequently, the predictive capacity of the model is likely constrained by the statistical relationships and parameter values derived from calibration sites that generally experienced less extreme conditions. This constraint systematically reduced the ability of the model to predict intense rainfall events, resulting in a tendency to underpredict extremes at validation sites where more severe events occurred.

**Empirical cumulative distribution function (ECDF) plots.** The ECDF plots of model errors demonstrated strong statistical agreement between generated and observed daily weather variables at both calibration and validation sites (Fig 7 for two representative sites; full results are provided in S1–S3 Figs).

Daily error distributions for Tmin and Tmax were generally well centered around zero, indicating unbiased temperature generation (Figs 7a,b and S1). Quantitatively, 59 ± 12% (mean ± 1 SD) of the daily Tmin and Tmax errors fell within ±1.0 °C, while 85 ± 10% were within ±2.0 °C across all sites. The ECDF curves for Tmax exhibited less steep gradients around zero error and displayed longer tails compared to Tmin, particularly evident at sites such as Pagoh, Paloh, Sitiawan, Temerloh, Alor Setar, Kluang, and Sandakan (Figs 7b and S1). This pattern indicates that the model's reproduction of Tmax distributions exhibited greater variability in error than that of Tmin distributions.

Daily wind speed error distributions demonstrated strong statistical agreement between the generated and observed values (Figs 7c,d and S2). The ECDF curves were characteristically steep and symmetric around 0 m s$^{-1}$, confirming

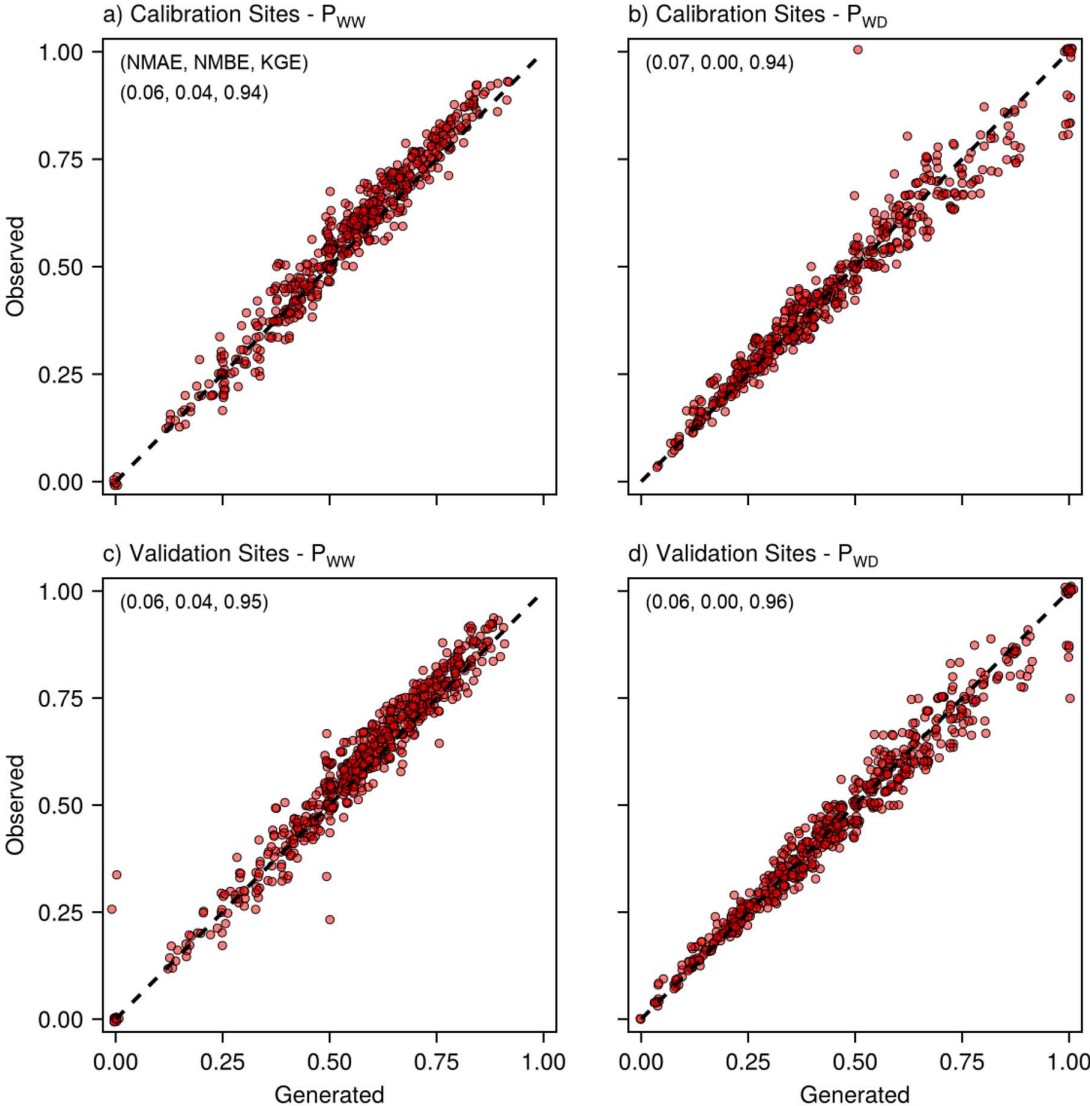

**Fig 4. Comparison of observed and model-generated monthly rainfall transition probabilities at calibration (a,b) and validation (c,d) sites** (n = 552 and 660 site-months, respectively). $P_{WW}$ **(a,c)** is the probability of a wet day following a wet day; $P_{WD}$ **(b,d)** is the probability of a wet day following a dry day. Each point represents one site-month. Dashed line indicates perfect agreement (1:1). Values in brackets are goodness-of-fit metrics Normalized Mean Absolute Error (NMAE), Normalized Mean Bias Error, and Kling-Gupta Efficiency (KGE).

very little systematic bias in wind speed generation. The model accurately captured the statistical properties of daily wind speed distributions, with 77 ± 13% (mean ± 1 SD) of errors within ±0.5 m s$^{-1}$ and 94 ± 6% within ±1.0 m s$^{-1}$ across all sites. Notably, validation sites generally exhibited larger error ranges than calibration sites, as exemplified by the broader error distributions at Kuala Terengganu, Melaka, and Sandakan, compared with the more constrained distributions at calibration sites, where only Kota Bharu showed extended tails (S2 Fig).

The ECDF curves for daily rainfall errors revealed approximately symmetric distributions centered on zero, confirming lack of systematic bias in the rainfall generation process (Fig 7e,f and S3). The steep gradients observed near zero error demonstrate that the model successfully captured the central tendencies of the observed rainfall distributions.

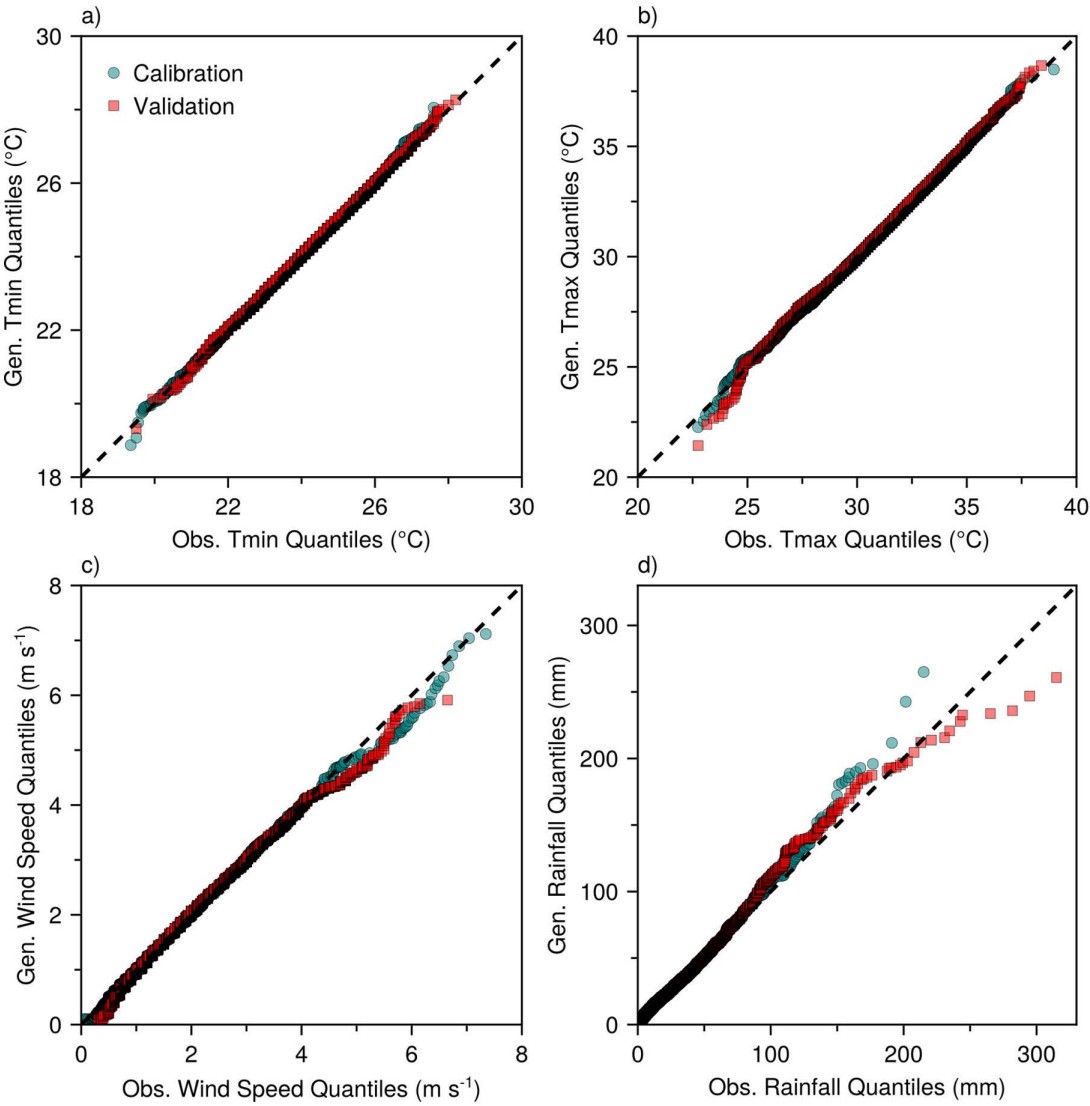

**Fig 5. Quantile-Quantile (QQ) plots comparing daily model-generated (gen.) and observed (obs.) Values for minimum air temperature (Tmin) (a), maximum air temperature (Tmax) (b), wind speed (c), and rainfall (d), shown separately for the calibration and validation sites.** Rainfall (d) includes wet days only. The dashed line indicates perfect agreement (1:1 line). Sample sizes for calibration and validation sites, respectively, are: n = 16797 and 20097 for Tmin, Tmax, and wind speed; n = 8389 and 10587 for rainfall.

Nevertheless, these curves exhibited more extended tails than the error distributions for temperature and wind speed, indicating that, while the model maintained overall distributional fidelity, larger errors occurred predominantly during the simulation of extreme rainfall events. This pattern reflects the inherent challenge of accurately reproducing the statistical characteristics of highly variable precipitation processes.

Quantitatively, rainfall generation demonstrated greater distributional variability than other meteorological variables. Across all sites, 58 ± 7% (mean ± 1 SD) of the daily rainfall errors fell within ±5 mm, 71 ± 7% within ±10 mm, and 83 ± 5% within ±20 mm. Consistent with the wind speed results, the validation sites generally exhibited larger rainfall error ranges than the calibration sites. The sites with particularly pronounced tail extensions were Kluang, Kuala Terengganu, and

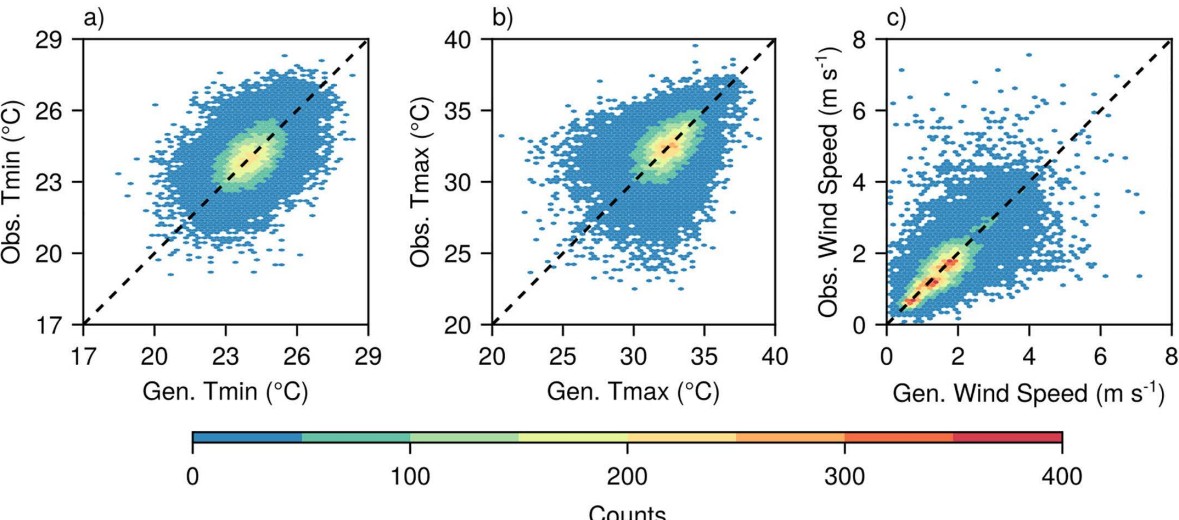

**Fig 6. Hexagonal bin plots showing agreement between observed (obs.) and model-generated (gen.) daily values of minimum air temperature (Tmin) (a), maximum air temperature (Tmax) (b), and wind speed (c).** The color intensity indicates the number of data points within each hexagonal bin, from low (blue) to high (red). The dashed line indicates perfect agreement (1:1 line). Data were combined from all calibration and validation sites (n = 36894).

Kuantan, all of which were validation sites (S3 Fig). This observation aligns with the previously noted finding that the validation sites experienced a higher mean annual rainfall than the calibration sites (2863 vs. 2267 mm; Table 1).

**Performance analysis across rainfall amount classes.** The observed rainfall distribution followed a typical tropical pattern, where light precipitation events predominated over heavy rainfall events [45,46]. In this study, 48.6% of days recorded no rain and 32.1% experienced light rainfall events (0.05–10 mm), while events exceeding 60 mm occurred in only 2.0% of observations (Fig 8a).

The accuracy of the model corresponded closely to the rainfall event frequency. No-rain days, constituting nearly half of all observations, achieved 75% of the predictions within ±5 mm error thresholds (Fig 8b), whereas light rainfall events maintained 60% precision at the same threshold. However, the model accuracy was reduced for larger but less frequent rain events. Moderate rainfall (10–30 mm) reached only 20% precision within ±5 mm, and heavy events (>30 mm) required tolerances exceeding ±50 mm to achieve comparable accuracy.

These limitations reflect the statistical challenge of modeling rare events from limited observational records [47] and the climatological differences between sites. For example, the validation sites experienced wetter conditions and higher rainfall extremes than the calibration sites, contributing to the divergence observed at the upper quantiles in the QQ analysis (Fig 5d) and the extended tails in the ECDF plots for some locations (S3 Fig).

While all rainfall classes eventually converged toward high precision at larger error thresholds, the no-rain and light rainfall classes demonstrated the steepest performance improvements. The extended tails for heavy events confirmed the model's ability to preserve central tendencies while struggling with the precise magnitudes of rare extremes, which is a well-documented limitation of stochastic weather generation [47,48]. Nonetheless, the model's ability to accurately simulate common rainfall patterns, which make up over 80% of the observations, provides valuable practical benefits for assessing impacts and conducting long-term climate studies.

## Oil palm yield simulations

The generated daily Tmin and Tmax (Fig 9a–d) closely aligned with the observed values along the identity line across the full range of quantiles, with only minor deviations occurring at the extreme tails. This pattern mirrors the performance

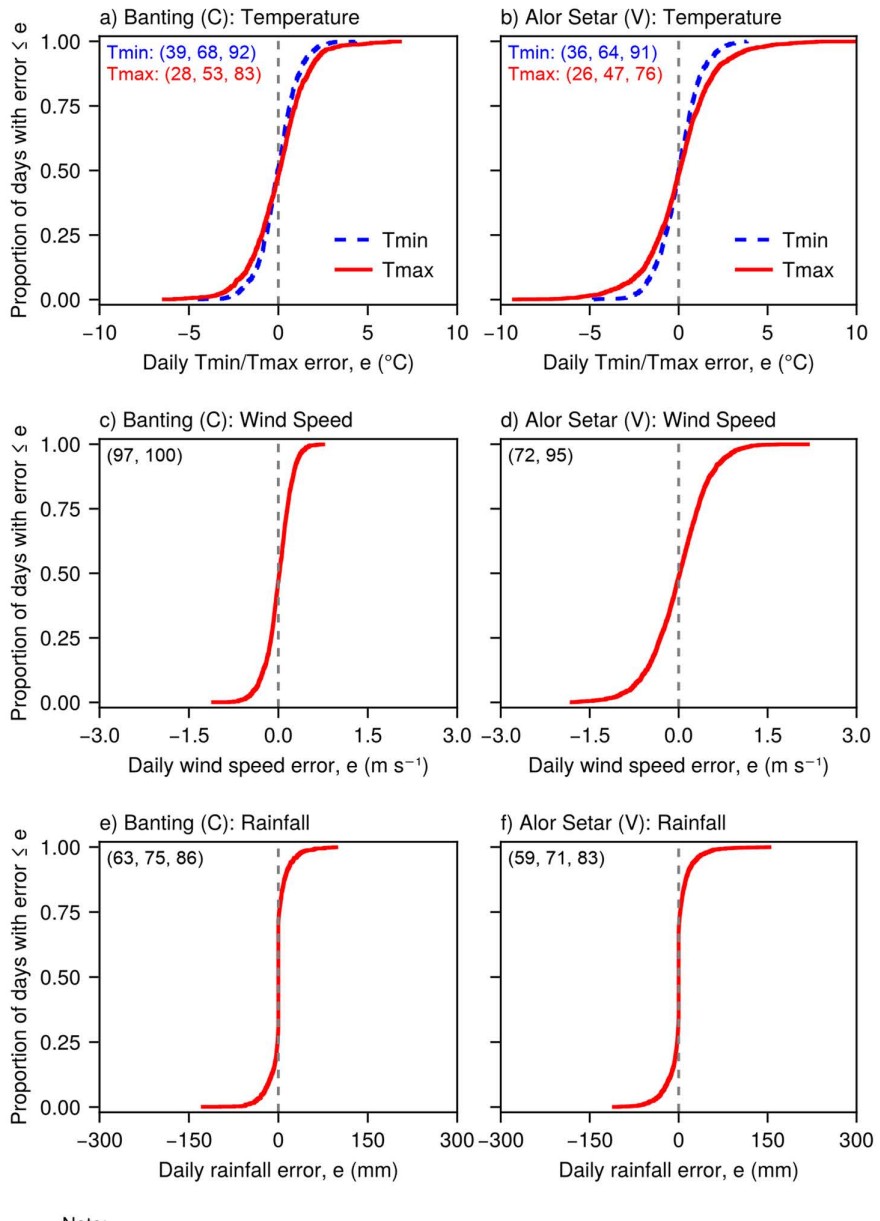

Note:
For temperature panels, values in brackets show % of days with |error| ≤ 0.5, 1.0, and 2.0 °C.
For wind speed, values in brackets show % with |error| ≤ 0.5 and 1.0 m s⁻¹.
For rainfall, values in brackets show % with |error| ≤ 5, 10, and 20 mm.

**Fig 7. Empirical cumulative distribution function (ECDF) plots of daily weather generation errors at a representative calibration site (Banting, left) and validation site (Alor Setar, right).** Panels show (a,b) minimum (Tmin, blue dashed) and maximum (Tmax, red solid) air temperature, (c,d) wind speed, and (e,f) rainfall. Vertical dashed lines indicate zero error. Values in brackets show the percentage of days within specified error thresholds. Sample sizes for each site are provided in Table 1. Complete ECDF results for all 23 sites are in S1–S3 Figs.

observed at the calibration and validation sites, confirming the transferability of the model to new locations. Wind speed generation (Fig 9e, f) showed strong agreement for values up to approximately 3 m s⁻¹ (similar to Fig 6c), beyond which the model exhibited the characteristic underestimation of higher wind speeds that was consistently observed across all

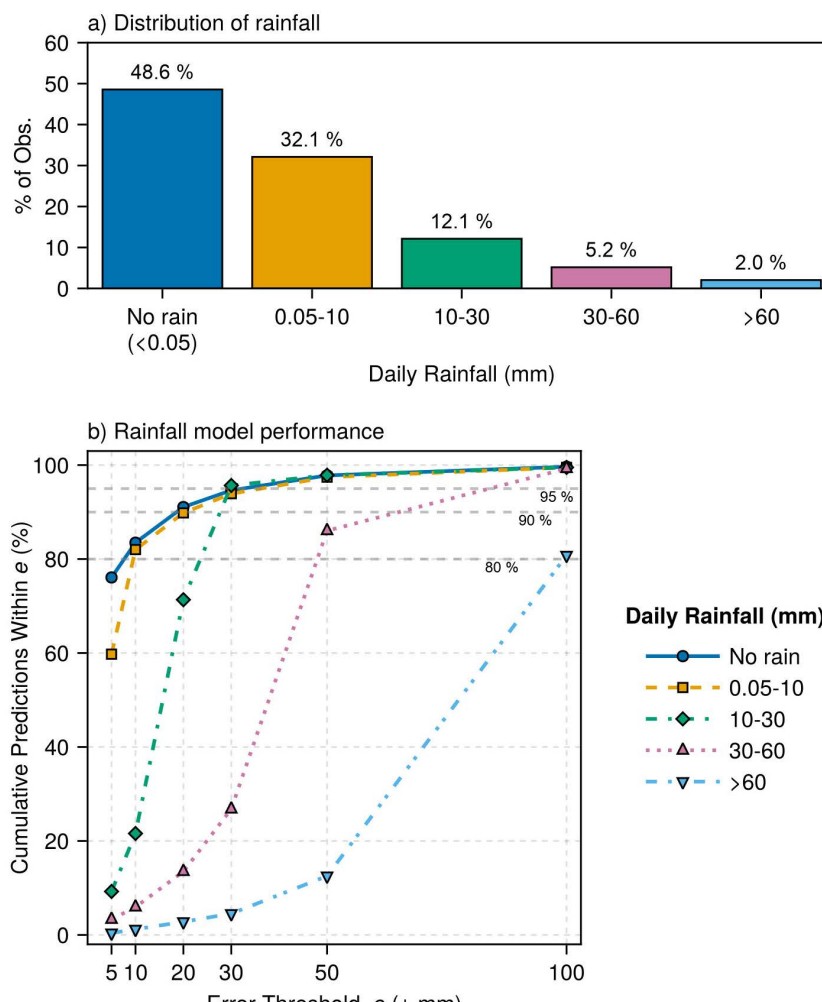

**Fig 8. Distribution of observed daily rainfall classes (a) and cumulative percentage of model-generated values within increasing error thresholds for each rainfall class (b).** Data were aggregated from all calibration and validation sites (n = 18976).

sites. The rainfall generation at both sites (Fig 9g–h) captured lower and moderate precipitation amounts effectively, but showed expected limitations in reproducing the most extreme events above 80–100 mm, consistent with the earlier broader site analysis (Fig 5d).

The ECDF plots provide additional confirmation of the model's unbiased performance at the Kerayong (Fig 10a, c, e, g) and Kalumpong (Fig 10b, d, f, h) sites. The error distributions for all meteorological variables were centered around zero, indicating very little systematic bias in the generated weather sequences. Quantitatively, 46–81% of daily temperature errors were within ±1 °C (Fig 10a–d), while wind speed errors showed even tighter distributions with at least 90% of values within ±1 m s$^{-1}$ (Fig 10e, f). More than two-thirds of the rainfall errors were within ±10 mm (Fig 10g, h). These precision levels reinforce the model's consistent performance across different geographical locations within the country.

Generated weather at Kerayong and Kalumpong over 20 year periods (Figs 9 and 10) produced trends similar to those observed at calibration and validation sites: good agreement for common conditions but underestimation of extreme rainfall (>80–100 mm) and wind speeds (>3–4 m s$^{-1}$) (compare Figs 9e–h with 5c, d).

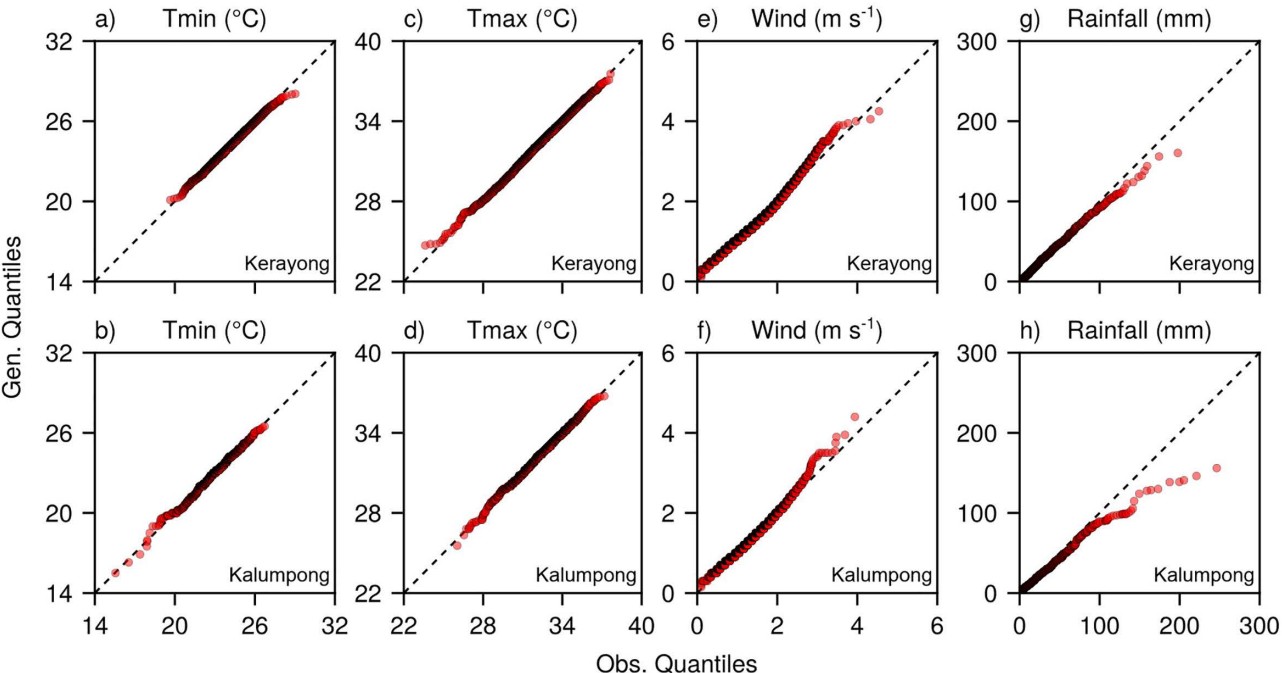

**Fig 9. Quantile-Quantile (QQ) plots comparing daily minimum air temperature (Tmin) (a, b), maximum air temperature (Tmax) (c, d), wind speed (e, f), and rainfall (g, h) between model-generated (gen.) and observed (obs.) values for the Kerayong (first row) and Kalumpong (second row) sites.** The rainfall (g, h) included only wet days. The dashed line indicates perfect agreement between the generated and observed values (1:1 line). Sample sizes for Kerayong and Kalumpong sites, respectively, are: n = 8035 and 8400 for Tmin, Tmax, and wind speed; n = 4234 and 2779 for rainfall.

The Sawit.jl oil palm model, driven by both generated and observed weather data, was used to evaluate the reliability of the weather generation process. If synthetic weather accurately represents real weather conditions, the resulting yield simulations should exhibit strong agreement with each other. This was indeed observed at both Kerayong (Fig 11a–c) and Kalumpong (Fig 11d–f) sites, where simulations using generated weather (red dashed lines) closely followed those based on observed weather (blue solid lines) across all planting densities. Importantly, the generated weather preserved the key temporal dynamics and yield magnitudes of fresh fruit bunch (FFB) production, with both approaches successfully replicating the characteristic interannual variability of oil palm systems.

**Rainfall intensity distribution effects.** Oil palm yield responses to rainfall manipulation varied systematically with the magnitude and type of intensity redistribution (Fig 12).

Scenario Set 1 was the progressive reduction of higher-intensity rainfall. Removing higher-intensity rainfall (H, > 60 mm) without redistribution produced strong linear yield declines at both sites (Fig 12a,b). Yields decreased progressively from approximately 5–7% at 25% H removal to 25–45% at complete H removal, with response curves nearly parallel across planting densities.

Scenario Set 2 was the progressive redistribution from higher- to lower-intensity rainfall (H→L). When removed H rainfall was redistributed to lower-intensity (L, ≤ 60 mm) events to conserve total rainfall, yield impacts were minimal to slightly positive (Fig 12c,d). Yields increased by approximately 2.5% at Kerayong and 1–8% at Kalumpong relative to the baseline, demonstrating that redistributing rainfall from higher- to lower-intensity events can enhance yields even when total rainfall remains constant.

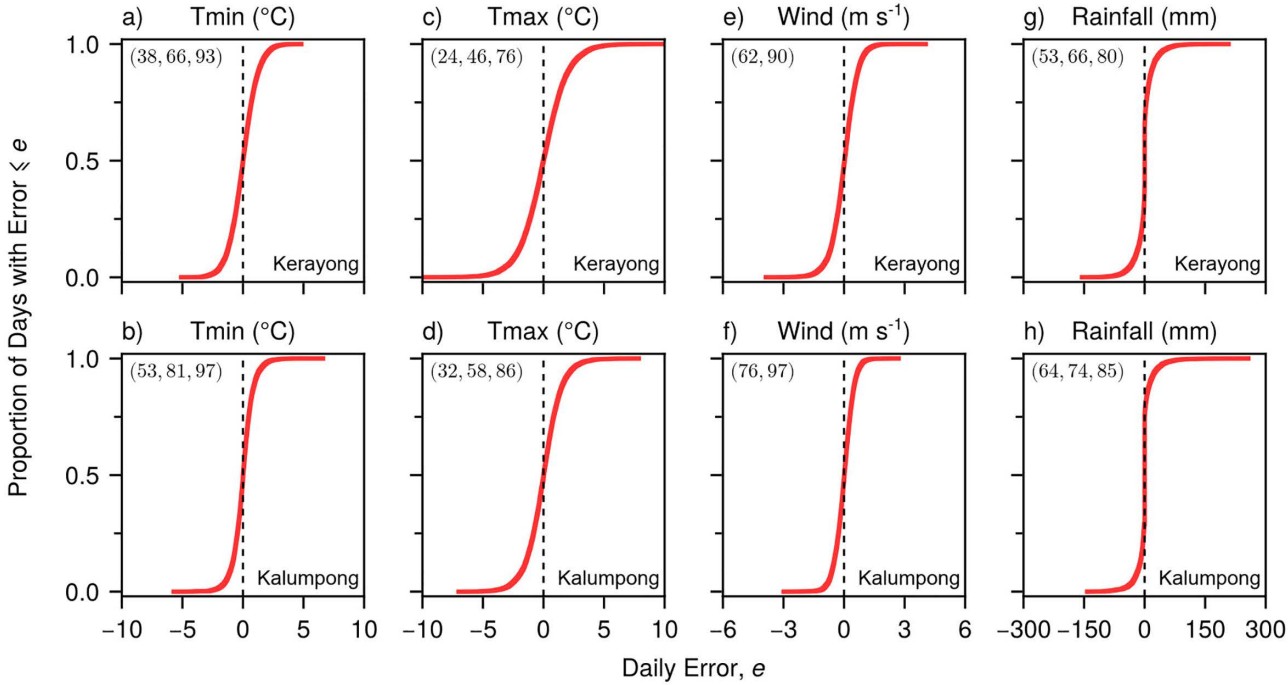

**Fig 10. Empirical cumulative distribution function (ECDF) plots comparing observed and model-generated daily minimum air temperature (Tmin) (a, b), maximum air temperature (Tmax) (c, d), wind speed (e, f), and rainfall (g, h) between model-generated and observed values for the Kerayong (first row) and Kalumpong (second row) sites.** Values in parentheses at the top left of each panel give the percentage of generated weather values that are within successive error bands. For Tmin and Tmax, the bands are $\pm 0.5$, 1, and 2 °C (a-d); for wind speed, they are $\pm 0.5$ and 1 m s$^{-1}$ (e-f); and for rainfall, they are $\pm 5$, 10, and 20 mm (g-h). The vertical dashed line indicates zero error. Sample sizes for Kerayong and Kalumpong sites, respectively, are: n = 8035 and 8400 for Tmin, Tmax, and wind speed; n = 4234 and 2779 for rainfall.

Scenario Set 3 comprised four comparative scenarios at 50%H manipulation (Fig 12e,f). Both deficit scenarios (L loss and H loss) produced substantial yield reductions of between 10 and 25% at both sites (pink background), with L loss causing greater declines than H loss despite equivalent rainfall removal. In contrast, the redistribution scenarios showed markedly different responses depending on transfer direction (green background). The H→L scenario produced minimal to positive yield changes (1–9%), while the L→H scenario resulted in yield declines of 5–15%.

These results demonstrate that maintaining total annual rainfall is more critical than preserving specific intensity distributions. However, oil palm yields showed greater sensitivity to reductions in lower-intensity rainfall than to equivalent reductions in higher-intensity rainfall, indicating that frequent, light and moderate rainfall events are more important for sustaining productivity than infrequent extreme events.

## Discussion

MsiaGen demonstrated consistent performance across the 23 sites spanning Peninsular and East Malaysia's tropical climate. At monthly scales, model performance is highly comparable across all 23 sites (NMAE <1.2% for air temperatures, <2.4% for wind speed, <1.8% for rainfall; KGE > 0.80; Fig 2). At daily scales, performance remains comparable for typical conditions (>90% of observations: temperatures 21–39°C, wind <3 m s$^{-1}$, rainfall <80 mm; Figs 5 and 6), but diverges for extremes (>95th percentile). Monthly-to-seasonal accuracy is more critical for crop modeling, as confirmed by successful oil palm yield simulations (Fig 11). The following sections discuss challenges in modeling different weather variables and evaluate the model's practical utility.

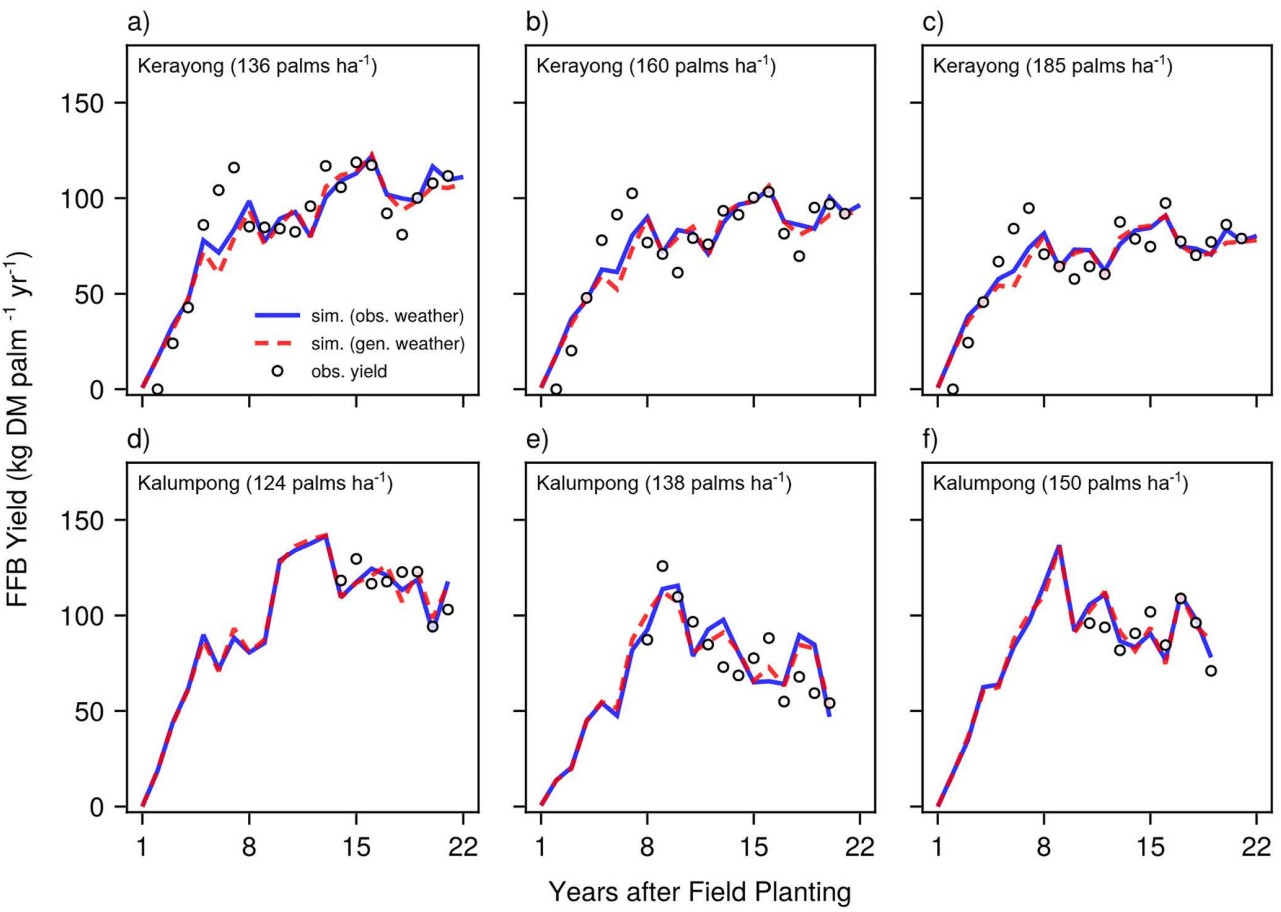

**Fig 11. Comparison of observed (obs.) (circles) and simulated (sim.) (red and blue lines) annual oil palm fresh fruit bunch (FFB) yield in kg dry matter (DM). Simulations at Kerayong (first row, a–c) were for planting densities of 136 (a), 160 (b), and 185 (c) palms ha⁻¹, whereas simulations at Kalumpong (second row, d–f) were for 124 (d), 138 (e), and 150 (f) palms ha⁻¹.** For each site, the simulated yields were driven by either the observed daily weather (solid blue line) or model-generated daily weather (red dashed line).

## Rain and wind speed generation problems and challenges

This study demonstrated that standard statistical models, specifically the Gamma distribution for rainfall and Weibull distribution for wind speed, are effective in modeling Malaysia's tropical climate patterns, while also confirming their known limitations in depicting extreme weather events.

Rainfall generation proved to be the most challenging aspect of weather simulation. At the monthly scale, the model demonstrated strong performance (low bias NMBE within ±2% and near-perfect overall agreement KGE > 0.95) (Fig 2m–o). However, the Anderson-Darling (AD) test revealed distributional discrepancies at 16–20% of sites (Fig 2p), which were higher than the 2–11% rejection rates observed for temperature and wind speed variables. This apparent contradiction between the excellent goodness-of-fit metrics and higher AD rejection rates, particularly for rainfall, is due to the test's greater sensitivity to tail deviations compared with other statistical measures. Although the AD test evaluates the entire distribution, it is more sensitive to tail deviations than other goodness-of-fit tests, such as the Kolmogorov-Smirnov test [43,49]. Consequently, some discrepancies in rainfall and wind speed extremes may be sufficient for the AD test to reject the null hypothesis of distributional equality, even when the model accurately captures the more frequent conditions that

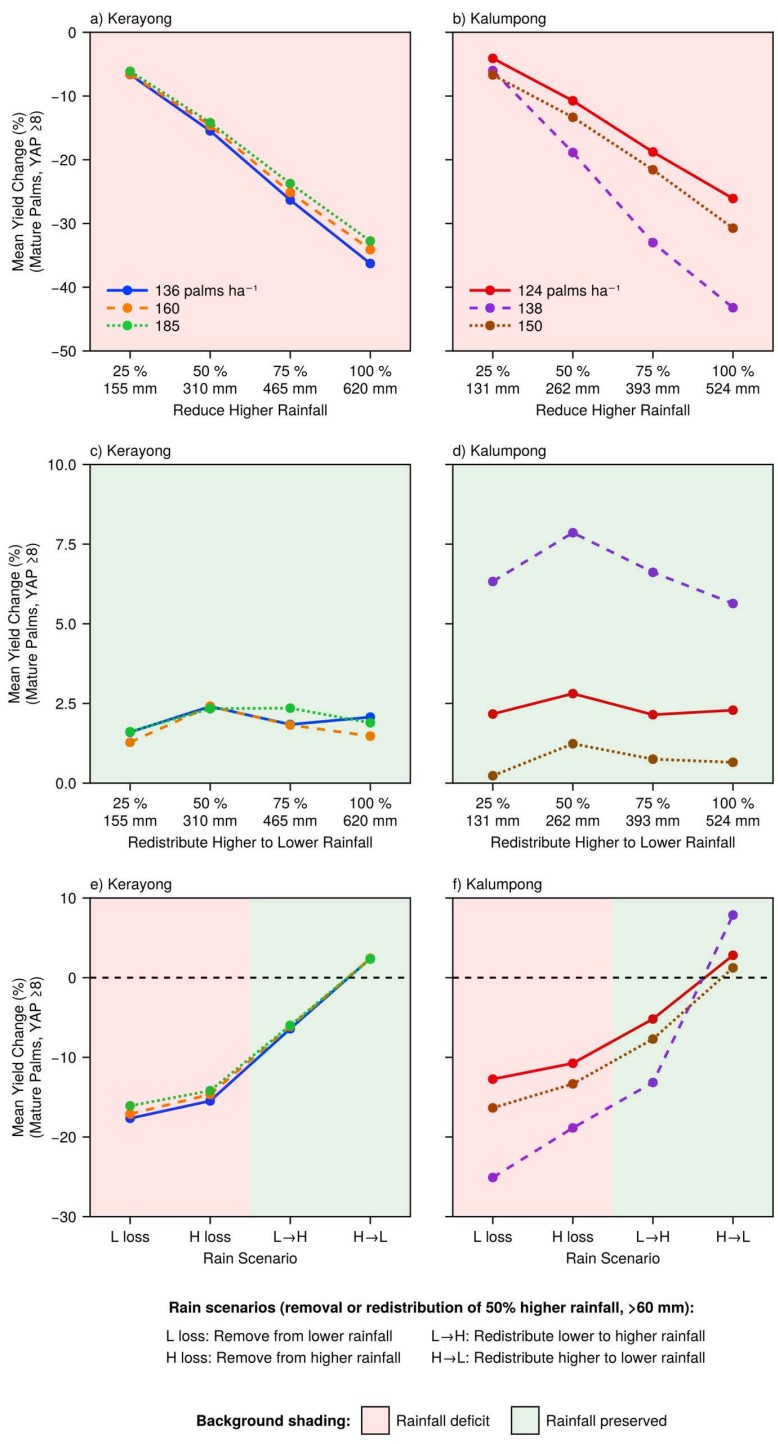

**Fig 12. Oil palm mean yield responses to rainfall intensity manipulations at Kerayong (left) and Kalumpong (right) sites for different planting densities.** Panels show (a,b) graduated reduction of higher-intensity (H) rainfall, (c,d) graduated H-to-lower-intensity (L) redistribution, and (e,f) four scenarios manipulating 50% of H rainfall: L loss, H loss, L→H, and H→L. Pink background indicates rainfall deficit; green indicates rainfall was conserved. Note: YAP is the number of years after field planting.

dominate both datasets. This pattern reflects the inherent complexity of tropical climatic processes, where intense convective events, highly variable precipitation patterns, and dynamic wind regimes driven by monsoons and diurnal cycles create statistical distributions that are difficult to capture using standard parametric approaches.

The accuracy of the model showed a clear inverse relationship with the magnitude of meteorological extremes. For daily rainfall, the accuracy achieved 75% of predictions within ±5 mm for no-rain days and 60% for light events (<10 mm), but declined to only 20% precision for moderate events (10–30 mm) and required error tolerances exceeding ±50 mm for heavy rainfall (>30 mm) (Fig 8b). The ECDF curves for daily rainfall errors further revealed near-symmetric distributions centered on zero with steep gradients near zero, confirming unbiased rainfall generation and the successful capture of central tendencies (S3 Fig). However, these curves exhibited extended tails, which is evidence of the deteriorating performance of the model during extreme events. This pattern was corroborated by QQ plots showing a good distributional fit for lower and moderate rainfall amounts, but increasingly large deviations from the identity line above 100 mm (Fig 5d). Similarly, wind speed generation demonstrated a good representation of typical (dominant) conditions, only beginning to show discrepancies at extreme wind speeds (Figs 5c, 6c and S2).

These performance gradients reflect the statistical challenge of modeling rare and extreme events [37,46]. Light rainfall events and moderate wind speeds are predominantly associated with stable atmospheric conditions; therefore, they exhibit more predictable statistical behavior. In contrast, heavy convective events and extreme wind episodes arise from complex interactions between atmospheric instability, moisture availability, topographic effects, and dynamic pressure systems, which are much more difficult to capture through purely statistical approaches.

Furthermore, MsiaGen was calibrated using only a short period (3–5 years) of daily weather data. Semenov and Barrow [10] recommended that historical records spanning 20–30 years or more are generally preferred to derive reliable and stable parameters for statistical distributions in such models. As noted earlier, the data lengths in this study were standardized by truncating longer available records to ensure spatial equity across the multisite network, thereby avoiding bias from any one station. Because MsiaGen was designed for weather generation rather than climate trend detection, the focus was on preserving local statistical characteristics rather than ensuring full temporal homogeneity. Undetected inhomogeneities (such as station relocations or short-term anomalies) could bias monthly parameters, but formal homogenization tests were not applied because their power is limited for short series [30]. Nevertheless, the strong cross-site validation results and multidecadal oil palm simulations indicate that the extracted statistics are sufficiently representative for the intended agricultural applications. Calibration parameters derived from these 3–5 years records successfully generated weather for the 22–23 year periods at Kerayong and Kalumpong (Figs 9 and 10). They produced trends similar to those observed at calibration and validation sites (Figs 5, 7, and S1–S3). The results demonstrate that the parameters, calibrated using short-term records, exhibit sufficient temporal stability to be reliably applied in long-term agricultural modeling and scenario analysis.

However, having temporal stability does not resolve the issue of inadequate extreme event sampling. The short calibration period provided insufficient observations of rare events (e.g., only 2% of days exceeded 60 mm rainfall; Fig 8a) that are needed to accurately characterize distribution tails. This had likely led to underestimation of extremes, but despite this limitation, the model remains suitable for oil palm applications where cumulative seasonal patterns, which MsiaGen accurately captures, drive yield responses more than rare extreme events (Fig 11).

Another possible limitation is the temporal sequencing of wet and dry days. The first-order Markov chain, as used in MsiaGen, although simple and fast to parametrize [31,37], may not fully capture the persistence structures associated with Malaysia's monsoons. Kemsley et al. [50] found that while the Bayesian Information Criterion (BIC) selected a first-order model for 56.2% of tropical grid cells, the third-order model was superior in capturing the wet/dry spell lengths and the interannual variability of precipitation occurrence in the tropics. The optimal Markov order for Malaysia has seldom been identified. Deni et al. [51], using 21–35 year records from 18 stations in Malaysia, concluded that first-order dependence was optimal for all monsoon seasons, but their decision criterion was BIC only. As pointed out by Kemsley et al. [50], BIC's parsimony penalty can mask deficiencies in reproducing wet/dry spells and interannual variability.

Quantitative validation of wet/dry spell length distributions reveals both strengths and limitations of the first-order Markov chain (S4 and S5 Figs). At the aggregate level, when pooling spell lengths across all sites, the model successfully reproduced the overall frequency distributions, with ECDF plots showing close alignment between observed and generated sequences throughout the common range (S4 Fig). Box plot comparisons confirm similar medians, interquartile ranges, and distributional spreads for both wet and dry spells (S4 Fig).

However, site-level analysis reveals systematic biases (S5 Fig). Sites experiencing longer wet spells showed consistent underestimation, while sites with longer dry spells showed a small tendency for overestimation. Maximum dry spell lengths were captured well, but maximum wet spell lengths were systematically underestimated across most sites, consistent with the broader limitation in extreme event representation discussed earlier. This pattern indicates that while the model captured the typical range of persistence well, it tended to dampen extreme persistence events, a recognized limitation of first-order Markov chains [50].

However, adopting higher-order Markov chains to address these limitations is impractical due to their rapidly increasing data requirements. A $k$-th order wet/dry chain requires $2^k$ conditional wet probabilities per month. Therefore, a first-order model needs only two probabilities: $P_{WW}$ and $P_{WD}$, but a second- and third-order model needs four and eight probabilities, respectively. Specifying these additional monthly transition probabilities is prohibitively complex for practical use.

At present, MsiaGen adequately captures typical dry spell durations that drive cumulative soil water depletion. While slight overestimation occurs at sites with longer dry spells, the successful yield simulations (Fig 11) demonstrate that this bias does not compromise the model's representation of water availability patterns relevant for oil palm productivity.

The Gamma distribution is widely used in rainfall modeling [4,6], but it suffers from well-documented limitations for extreme event simulations. The choice of Gamma distribution for rainfall in this study was motivated by its mathematical simplicity and parsimony of requiring a few parameters, yet providing adequate accuracy for a broad range of typical applications. However, rainfall studies in Malaysia highlight significant regional variability in distribution performance, suggesting that no single model universally fits all rainfall patterns. For instance, Syafrina et al. [16] compared Gamma and Weibull distributions across five stations in the dry northwest Peninsular Malaysia, finding that Weibull outperformed Gamma at some sites, while Gamma was better for others. Conversely, in the wet northeast Peninsular Malaysia, Gamma proved more suitable for all seven stations over a 30-year period [5,15], despite its lighter tail compared to Weibull's heavy-tailed shape, which is typically better for extremes. Another study [52] focused on three stations in Selangor (central-west), where Weibull was favored over Gamma, although these conclusions were limited by the small number of closely situated stations (≤120 km apart).

Broader comparative analyses repeat this spatial dependency. Tan et al. [53] evaluated ten stations (120–400 km apart) in Peninsular Malaysia using Exponential, GEV, Gamma, and Weibull distributions, identifying GEV as the best, but noting that Gamma and Weibull performed comparably and were close alternatives to GEV. Kong [54] found Gamma to be superior across five stations when tested against Weibull, Tweedie, Lognormal, and Pareto distributions. Conversely, Dan'azumi et al. [55] and Norzaida et al. [56] reported different results: the former favored Generalized Pareto for hourly intensities (12 stations, 10–20 years of data), while the latter advocated Mixed Exponential for clustered stations (<5 km apart). These inconsistencies underscore the complexity of rainfall modeling in Malaysia, where spatial and temporal heterogeneity (e.g., monsoonal influences and localized convection) are likely to drive distribution suitability [15,57].

Papalexiouy et al. [58] demonstrated that for modeling rainfall extremes, heavy-tailed distributions (such as Pareto, Lognormal, and Weibull) are better suited than light-tailed distributions (such as Gamma). They examined over 15,000 daily rainfall records, spanning 50–172 years, from various locations worldwide (though unfortunately did not include Malaysian data) and found that the heavy-tailed Pareto distribution was ranked the best fit to model rainfall extremes, followed by Lognormal and Weibull, while the light-tailed Gamma ranked the lowest. However, despite its poor average performance, the Gamma distribution was surprisingly the best fit for over a quarter (25.8%) of global rainfall records, outperforming the Weibull distribution. This seemingly paradoxical outcome is due to Gamma's binary-like performance,

where it was either the best or worst fit, with very few intermediate results. This volatility in Gamma's performance likely explains the mixed results observed across Malaysian studies and underscores the challenge of selecting appropriate distributions for diverse tropical rainfall regimes.

As discussed earlier, the model tends to underestimate wind speed extremes (>3–4 m s$^{-1}$) (Fig 5c). Wind speed sensitivity analysis at both oil palm simulation sites revealed site-dependent yield responses (S1 Table). At Kalumpong, reducing wind speeds by 20% increased yields by 0.25–4.40% across the three planting densities, while increasing wind speeds by 20% decreased yields by 0.39–3.33% (S1 Table). At Kerayong, the responses were more pronounced across the three planting densities: reducing wind speeds by 20% increased yields by approximately 8%, while increasing wind speeds by 20% decreased yields by approximately 6% (S1 Table). These responses reflect wind speed's influence on evapotranspiration (ET) dynamics, where higher wind speeds increase atmospheric water demand and accelerate soil moisture depletion. It is, however, radiation and temperature that are the primary drivers of ET, not wind speed. Particularly in humid, tropical climates, wind speed has a small effect on evapotranspiration [59,60]. Fisher et al. [59] calculated that wind speed only explained 4% of the variance in monthly evapotranspiration in the tropics, while net radiation dominated with 87% of variance explained. Wind speed affects yields indirectly through evaporative demand, contrasting with rainfall's direct control of soil water supply. While wind-induced yield changes can reach 8%, they remain smaller than the 5–45% yield responses to rainfall manipulations (Fig 12a,b), with rainfall availability being the dominant driver of oil palm productivity in Malaysia's tropical environment.

The model's underestimation of extreme wind speeds has small practical impact for three reasons. First, extreme wind events are rare at both oil palm simulation sites: the 95th percentile wind speeds were 2.4 and 2.0 m s$^{-1}$ at Kerayong and Kalumpong, respectively, indicating that wind speeds >3 m s$^{-1}$ occur on fewer than 5% of days, with speeds >4 m s$^{-1}$ being even rarer (less than 0.1% of days). Second, the ±20% wind scenarios represent a uniform perturbation affecting all days annually, whereas the model underestimates wind speeds only on the rare days when winds exceed 3–4 m s$^{-1}$. The actual yield impact from underestimating wind on these infrequent extreme days would therefore be substantially smaller than the 6–8% observed with year-round ±20% changes. Third, the close agreement between yield simulations using generated versus observed weather at both sites (Fig 11) empirically confirms that this limitation does not compromise the model's utility for oil palm yield prediction.

Standard distributions struggle to model extreme rainfall and wind speed events in Malaysia's dynamic tropical climate, where time-varying statistical properties arise from seasonal shifts in the Inter-Tropical Convergence Zone (ITCZ), monsoon patterns (northeast from November to March), and El Niño-Southern Oscillation (ENSO) variability [29]. These effects violate the stationarity assumptions inherent to standard distributions. For instance, analysis of the results from Papalexiou and Koutsoyiannis [61], drawing on a global dataset of 15,137 daily rainfall records spanning 40–163 years, shows that lighter-tailed distributions like the Gumbel consistently underrepresent precipitation extremes, particularly for return periods beyond 100 years. This results in approximately 20–50% underestimations of extreme rainfall intensities globally and up to 60–70% in tropical regions, like Southeast Asia, which are driven by intense convective systems.

Furthermore, tropical datasets that exhibit heavy-tailed and multimodal behaviors from convective events and diurnal cycles violate the lighter-tailed or unimodal assumptions of standard models. These standard distributions also neglect spatial-temporal dependencies, including orographic rainfall differences between Peninsular and East Malaysia, as well as persistent wet spells, potentially introducing 10–20% errors in regional estimates [51,62].

Recent efforts have attempted to address these limitations of lighter-tailed distributions using hybrid modeling techniques. These include mixed Gamma-Generalized Pareto models for improved tail representation [63] and bivariate Gamma mixture distributions to capture multimodal rainfall patterns [64]. However, such approaches significantly increase model complexity, often requiring several additional parameters beyond the standard distribution, which may outweigh the modest gains in extreme event modeling for practical agricultural applications.

Furthermore, for tropical agricultural applications, particularly perennial cropping systems such as oil palm, the strength of the current implementation in capturing seasonal rainfall accumulation patterns may be more relevant than precise extreme event simulations. The results of this study (Fig 11) suggest that oil palm yield responses depend more heavily on seasonal totals (which the model effectively captures) than on infrequent extreme events. Despite its first-order Markov chain, MsiaGen performs well for oil palm because of the crop's response to accumulated weather rather than extreme short-term events. This lends support to the continued use of simple distributions for agricultural weather generation, while acknowledging the need for enhanced approaches in applications requiring precise extreme event characterization, such as flood risk assessment.

**Comparative evaluation of alternative distributions.** Given these documented limitations and the regional variability in distribution performance, a post-hoc comparison was conducted between the Gamma distribution and two alternatives: the Generalized Pareto Distribution (GPD) and Mixed Exponential (MixExp) distribution. GPD was selected based on its documented superiority for modeling rainfall extremes [53,58], while MixExp has shown promise in Malaysian contexts [56]. Both these alternative distributions require three parameters, one more than the two-parameter Gamma distribution. This additional parameter grants them greater flexibility, enabling a more accurate representation of complex distributional features in the data. Specifically, the GPD is defined by location, scale, and shape parameters, while the MixExp uses a mixing weight and two distinct rate parameters.

To evaluate whether alternative distributions could justify the added complexity, Gamma was fitted using method of moments (MoM), which is the required approach for MsiaGen, while GPD and MixExp were fitted using maximum likelihood estimation (MLE), which is their optimal parameter estimation method. This comparison deliberately favors the alternatives by allowing them their best-case performance while constraining Gamma to MsiaGen's operational requirements. If GPD or MixExp cannot substantially outperform MoM-fitted Gamma even when using MLE, they offer no practical advantage for MsiaGen.

Comparative analysis across all 23 sites revealed that even with the advantage of MLE optimization, neither alternative consistently outperformed Gamma (Table 2). Gamma (MoM) achieved the lowest RMSE at 14 sites (61%), while MixExp (MLE) performed best at 9 sites (39%), and GPD (MLE) showed consistently poor fit across all locations. However, even at sites where MixExp outperformed Gamma, the RMSE differences were generally modest (mean absolute difference of 0.24 mm, representing 17% of mean Gamma RMSE, excluding one outlier at Paloh), whereas GPD underperformed by an average of 1921% across all sites. GPD's poor performance confirms its unsuitability for modeling complete rainfall distributions, as it is fundamentally designed for threshold exceedances rather than the full range of rainfall values. The modest advantage of MixExp at one-third of sites, despite using MLE, does not justify abandoning the simpler MoM-based parameterization, particularly given the operational requirements discussed below.

MsiaGen's architecture requires MoM for parameter estimation, which provides critical advantages beyond distributional fit. MoM derives parameters directly from observable statistics (such as mean and variance) that users can calculate from local weather data or climatological databases. Gamma's closed-form MoM solutions, such as the shape parameter $s = (mean/SD)^2$ and scale parameter $\theta = SD^2/mean$, enable practitioners to parameterize and adjust the model without specialized statistical software. In addition, these parameters have direct physical meaning: shape relates to rainfall variability and scale to rainfall intensity, allowing users to input estimates based on local knowledge or projected climate scenarios.

In contrast, neither GPD nor MixExp can be reliably parameterized using MoM, requiring instead MLE through iterative numerical optimization. While MLE can provide better fits at some locations (as demonstrated in Table 2), it requires computationally intensive algorithms that are impractical for multi-site generation, sensitive to initial parameter values, prone to convergence failures, and produces parameters without intuitive physical interpretation (unlike Gamma). This would make the weather generator significantly less accessible to practitioners who lack programming expertise or statistical software, and would prevent transparent parameter adjustment because MLE-derived parameters cannot be easily recalculated or

**Table 2. Root mean square error (RMSE, in mm) comparison of parametric distributions fitted to observed daily rainfall across all 23 sites. RMSE is calculated from differences between empirical and fitted quantiles across 100 percentiles. Gamma was fitted using method of moments (MoM) as required by MsiaGen, while Generalized Pareto Distribution (GPD) and Mixed Exponential (MixExp) were fitted using maximum likelihood estimation (MLE) to evaluate their best-case performance. Lower RMSE values indicate closer agreement between observed and predicted rainfall distributions.**

| Site | Gamma (MoM) | GPD (MLE) | MixExp (MLE) | Best fit |
|---|---|---|---|---|
| *Calibration sites* | | | | |
| Lubok Merbau | 0.84 | 14.55 | 1.06 | Gamma |
| Sitiawan | 1.37 | 18.74 | 0.90 | MixExp |
| Temerloh | 1.38 | 18.29 | 1.08 | MixExp |
| Serdang | 0.62 | 19.94 | 2.15 | Gamma |
| Banting | 1.20 | 18.30 | 1.13 | MixExp |
| Kuala Pilah | 0.67 | 18.62 | 0.89 | Gamma |
| Pagoh | 0.67 | 18.58 | 0.69 | Gamma |
| Paloh | 5.83 | 12.83 | 3.63 | MixExp |
| Layang Layang | 0.82 | 20.64 | 1.03 | Gamma |
| Kota Bharu | 1.41 | 27.02 | 2.86 | Gamma |
| Sibu | 1.12 | 25.46 | 2.34 | Gamma |
| Tawau | 0.88 | 16.33 | 0.84 | MixExp |
| *Validation sites* | | | | |
| Alor-Setar | 0.63 | 19.72 | 0.41 | MixExp |
| Teluk-Intan | 1.13 | 18.02 | 0.88 | MixExp |
| Melaka | 0.69 | 19.41 | 1.18 | Gamma |
| Kluang | 1.52 | 21.81 | 1.66 | Gamma |
| Kuantan | 2.18 | 21.23 | 2.04 | MixExp |
| Kuala Terengganu | 3.14 | 29.81 | 3.84 | Gamma |
| Kuching | 1.33 | 25.79 | 1.48 | Gamma |
| Bintulu | 1.64 | 26.57 | 3.15 | Gamma |
| Miri | 1.22 | 23.78 | 0.79 | MixExp |
| Kota Kinabalu | 0.76 | 23.79 | 1.51 | Gamma |
| Sandakan | 0.72 | 24.77 | 0.97 | Gamma |

modified by users when exploring climate scenarios or incorporating local expert knowledge. The opaque nature of iteratively-optimized parameters removes user agency in model calibration and validation.

Most importantly, despite Gamma's suboptimal fit at some locations and its limitations in capturing extreme events, oil palm yield simulations using Gamma-generated weather demonstrated strong agreement with observed yields (Fig 11). This confirms that Gamma's representation of seasonal rainfall patterns is sufficient for the intended agricultural application, where seasonal totals matter more than individual extreme events. The selection of Gamma balances adequate statistical performance with operational requirements: closed-form parameterization, physical parameter interpretability, computational efficiency, and user accessibility.

## Air temperature generation

Compared with rainfall and wind speed, the generated air temperature (Tmin and Tmax) data were the most accurate. MsiaGen demonstrated strong performance in reproducing air temperature patterns across both temporal scales. At

the monthly level, the model achieved high accuracy, with mean absolute errors below 1.2%, very low bias, and strong overall agreement scores (KGE > 0.8) at most sites for both Tmin and Tmax (Fig 2a–h). The distributional accuracy of the model was robust, with low AD test rejection rates (<2.2%) for both temperature variables. Daily temperature reproduction showed a similarly strong performance, with QQ plots revealing tight clustering along the identity line, particularly within the common temperature ranges of 21–28 °C for Tmin and 25–39 °C for Tmax (Fig 5a–b). Although the model exhibited minor deviations at temperature extremes, especially for values below these ranges, the overall distributional accuracy remained high. The ECDF plots confirmed little systematic bias, with well-centered error distributions around zero (S1 Fig). Notably, 59±12% of the daily temperature errors fell within ±1.0 °C, and 85±10% were within ±2.0 °C across all sites. The model showed slightly greater variability in reproducing Tmax than Tmin, with Tmax errors displaying broader distributions and longer tails.

This strong performance in temperature generation can be attributed to Malaysia's inherently stable temperature regime. The country's equatorial position (1–6 °N) produces a highly stable temperature regime with little annual variation. Daily historical records of Peninsular Malaysia (1991–2020) indicate typical daytime maxima between 30 and 33 °C and nighttime minima between 23 and 25 °C [65]. The equatorial location and maritime surroundings of Malaysia create nearly constant solar insolation and moderate temperature extremes, resulting in narrow seasonal and diurnal temperature ranges. This climate stability makes temperature patterns highly predictable, enabling MsiaGen to accurately reproduce them.

Daily air temperature is traditionally approximated using the Normal distribution [33]. Older established WGs such as USCLIMATE, WXGEN, LARS-WG, CLIMGEN, and CLIGEN all use the Normal distribution to generate Tmin and Tmax. Harmel et al. [66], however, cautioned that air temperatures in the US (1961–1990) often do not follow Normal distribution and are skewed for some months. They discovered that modeling air temperatures using this distribution could lead to improbable values.

In this study, nearly all calibrated sites exhibited moderate skewness in their temperature distributions, especially for Tmax, with sites typically showing such skewness for one to five months per year on average. As a result, the Skew Normal (SN) distribution [35] was chosen for modeling these data. SN is advantageous for its mathematical simplicity and parsimony. It offers closed-form expressions for moments like the mean and variance and straightforward mappings (relationships) between parameters and these moments, all of which make parametrizing SN tractable. An additional benefit is that SN reduces to the standard Normal distribution when the skewness parameter is set to zero.

This study appears to be the first to apply the SN distribution to model air temperatures in Malaysia. Previous research have predominantly relied on the GEV distribution to capture the asymmetry of Malaysia's air temperatures [67–69], although Supian and Hasan [70] recently demonstrated the superiority of the Generalized Skew Logistic (GSL) distribution. In their analysis of maximum temperatures (1994–2017) across 17 Malaysian stations, the GSL provided the best fit for 71% of stations, outperforming the standard Normal, Lognormal, Gamma, and Weibull distributions (where the Normal distribution was optimal for only one station).

Similar to SN, the GSL distribution is defined by three parameters but adds an optional fourth, a shape parameter, to enhance flexibility, all of which enable GSL to better capture heavy tails and extreme skewness. However, unlike SN (which reduces to a standard Normal distribution when symmetric), GSL maintains heavier tails even without skewness, making it more adaptable for extreme deviations.

Despite these advantages, GSL often lacks closed-form moment solutions for its parameters [71], thus complicating its parameter estimation. Given SN's strong performance in this study and its computational simplicity, SN is better suited (and more practical) for modeling moderate skewness in air temperatures. However, GSL flexibility would be useful in cases requiring extreme asymmetry.

### Weather generation for oil palm modeling

Oil palm FFB yield simulations using generated and observed weather (Fig 11) pose a stringent test of the MsiaGen's practical utility, as they integrate the cumulative effects of all meteorological variables over multidecade simulation

periods. Oil palm yield simulations at both Kerayong and Kalumpong sites (Fig 11) showed remarkably close agreement between simulations driven by the generated and observed weather. Moreover, the generated weather maintained the essential temporal patterns and magnitudes of FFB production, with both simulation approaches capturing the interannual variability that characterizes oil palm production systems.

This close correspondence demonstrates that MsiaGen successfully preserved not only the statistical properties of individual meteorological variables but also key inter-variable correlations (Fig 3) and temporal dependencies (Fig 4) that drive crop physiological responses. Weather generation for over 20 years period at both Kerayong and Kalumpong plantation sites using parameters derived from short-term records confirms temporal stability of the parameters.

The consistency of the model performance across different planting densities is particularly noteworthy, as higher densities typically create more competitive growing conditions and greater sensitivity to weather variability [72,73], yet the generated weather continued to produce realistic yield patterns even under these demanding scenarios.

The nearly identical results from the two yield simulations, despite MsiaGen not capturing every extreme weather event, underscore an important physiological aspect of oil palm: the formation of oil palm bunches is influenced by the cumulative balance of weather properties (such as solar radiation, temperature, and water supply) over extended periods rather than by sporadic heat waves or heavy rainfall [24,25,74]. In Malaysia's tropical climate, the cumulative effects of weather conditions, which MsiaGen accurately models, are vital for shaping oil palm production. MsiaGen effectively supports long-term agricultural planning and risk assessment for perennial crops, particularly when the historical weather data are insufficient.

Within Malaysia's tropical climate, mean daily wind speeds above $4 \, m \, s^{-1}$ are rare [75,76]. In this study, rainfall events exceeding 30 mm constituted merely 7% of all observed precipitation across the 23 sites, with the most extreme events (>60 mm) representing only 2% of observations (Fig 8a). Specifically at Kerayong and Kalumpong, rainfall exceeding 60 mm comprised 5% of rain days. MsiaGen accurately reproduced rainfall transition probabilities, both wet-following-wet ($P_{WW}$) and dry-following-wet ($P_{WD}$), at calibration and validation sites (Fig 4), indicating that the generator maintains correct rainfall persistence patterns and temporal sequencing. This preservation of rainfall frequency and timing is critical because oil palm production is determined principally by accumulated weather integrated over several months [74]. MsiaGen's robust performance in simulating the dominant 80% of common conditions, combined with accurate rainfall temporal patterns, proved sufficient to maintain realistic crop growth patterns over multidecade simulation periods. Additionally, the superior accuracy achieved in generating air temperature, a primary driver of crop metabolic processes, likely stabilized the overall yield predictions by providing accurate inputs for temperature-dependent functions within the crop model.

To address whether MsiaGen's underestimation of extreme rainfall events affects yield prediction accuracy, a systematic rainfall manipulation scenarios at Kerayong and Kalumpong was conducted (Fig 12). All deficit scenarios, whether removing from higher- or lower-intensity classes, produced yield declines, sometimes substantial, between 5 and 45% (Fig 12a,b), confirming that total water availability largely drives oil palm productivity. This relationship reflects oil palm's high and continuous water requirements throughout its growing cycle, with mature palms requiring approximately 4–5 mm day$^{-1}$ (15,000–40,000 m³ ha$^{-1}$ annually) [74]. However, when total rainfall was maintained through redistribution, yield responses varied depending on the direction of intensity transfer. This demonstrates that rain intensity distribution also influences productivity.

For instance, redistributing from higher- to lower-intensity events (H→L) in Scenario Set 2 increased yields by 1–8% despite unchanged total rainfall (Fig 12c,d). This enhancement occurs because lower-intensity events infiltrate more effectively into the soil profile, minimizing surface runoff losses that characterize extreme events while maintaining more consistent soil moisture availability. This pattern better matches the water demand of oil palm. The contrasting response in Scenario Set 3, where removing equivalent rainfall amounts from lower-intensity versus higher-intensity classes caused greater yield losses in the former case, further illustrates this differential sensitivity (Fig 12e,f). Lower-intensity events

provide frequent, regular moisture replenishment (occurring on 95% of rainy days and contributing 75–78% of annual rainfall), and their loss creates extended dry periods that would more easily stress palms.

The relatively modest impact of removing higher-intensity rainfall reflects several mechanisms. Extreme events often generate substantial runoff rather than plant-available soil moisture, and water from these events may percolate beyond the root zone. Moreover, because extreme events occur on only 5% of rainy days, their removal does not substantially disrupt the temporal pattern of water availability since frequent lower-intensity events continue providing regular moisture inputs. Together, these factors explain why oil palms are more sensitive to changes in frequent, lower-intensity rainfall than to equivalent changes in rare, extreme events.

These findings directly address concerns about MsiaGen's tendency to underestimate higher-intensity rainfall while accurately reproducing total annual rainfall. MsiaGen's behavior corresponds more closely to the H→L scenario in Set 3. The underestimation of extreme events does not markedly compromise yield prediction accuracy because frequent, light and moderate events (which MsiaGen accurately reproduces) dominate both rainfall contribution (75–78% of annual total) and yield determination. For oil palm yield modeling, accurate representation of rainfall frequency, temporal distribution, and total amount takes precedence over precisely capturing rare extreme event magnitudes.

The good agreement between the simulations supports MsiaGen's utility for long-term agricultural planning and risk assessment studies, where historical weather records may be insufficient. Its transferability to independent sites indicates that the model captures key climatological patterns generally representative of Malaysia's tropical environment. For perennial tropical commodities, whose productivity responds to slowly varying climate drivers, residual imperfections, particularly for rare extreme events, appear to have limited impact on yield projections. Consequently, the generator is suitable for scenario analyses that require large ensembles of internally consistent daily weather data, particularly for quantifying production risks under climate variability. However, further testing across a wider range of climatic conditions in the country and for other crop types is needed before broader claims of suitability for crop risk assessments can be made. Caution is also warranted for studies explicitly addressing damage processes linked to extreme storms or flood-triggering rainfall, as these conditions are still not fully represented by the current algorithm.

Although MsiaGen was developed specifically for Malaysia, its methodological principles are transferable to other equatorial Af climate regions. These principles include: distribution selection based on observed data properties, method-of-moments parameterization for accessibility, constraint compliance procedures, and multi-scale validation using statistical metrics and crop simulations. However, direct application of Malaysian parameters to other tropical regions would be inappropriate, as significant regional differences exist in monsoon characteristics, orographic rainfall enhancement, and maritime versus continental influences. Extension to other regions would require local recalibration following these principles. The MsiaGen framework thus provides a practical template for weather generator development in data-scarce tropical environments.

## Conclusion

The weather generator MsiaGen performed well in simulating air temperatures, with the Skew Normal distribution effectively capturing observed patterns, although with minor deviations at distribution extremes. For rainfall and wind speed, the model reliably reproduced central tendencies, successfully representing the light-to-moderate conditions that characterize approximately 80% of the daily weather records. However, performance declined for rare extreme events, particularly for daily rainfall exceeding 80–100 mm or wind speeds surpassing 3–4 m s$^{-1}$. These limitations are likely attributable to both the Gamma/Weibull tail properties and the short 3–5 year calibration period.

The accuracy of MsiaGen was further tested in oil palm yield simulations using two independent sites with 22–23 years of data, where results using generated weather data closely matched those from observed records. This long-term application confirmed temporal stability of model parameters while also demonstrating that the systematic underestimation of extremes (likely attributable to short calibration periods) persists consistently across decades. This close

agreement confirms that the generator adequately represents the weather patterns that most significantly influence oil palm productivity, namely the cumulative seasonal effects rather than rare extreme events. The successful yield simulations at both test sites, despite the model's limitations in extreme event representation, suggest that MsiaGen can serve as a valuable tool for oil palm system modeling where complete weather records are unavailable. However, model validation was limited to two sites and specific growing conditions, and further testing is needed to confirm broader applicability.

Potential model improvements should focus on increased representation of extreme events through alternative statistical approaches, although such refinements would need to balance increased accuracy against practical usability. Extended calibration periods would improve parameter estimation, and the model needs to be validated in more diverse agricultural systems. MsiaGen provides a functional solution for agricultural research applications in data-scarce tropical environments, particularly for studies of perennial crops where seasonal weather patterns outweigh the influence of transient extremes.

## Supporting information

**S1 File. Supplementary methods.**
(PDF)

**S1 Table. Simulated oil palm yield responses to ±20% daily wind speed manipulations at Kerayong (a) and Kalumpong (b) across three planting densities.** Values represent percentage change in annual fresh fruit bunch (FFB) yield relative to baseline simulations using unmodified observed wind speeds. YAP = years after planting. Negative values indicate yield declines; positive values indicate yield increases.
(PDF)

**S1 Fig. Empirical cumulative distribution function (ECDF) plots comparing observed and model-generated daily minimum air temperature (Tmin, blue dashed line) and maximum air temperature (Tmax, red solid line) at each calibration (C) site (a-l) and validation (V) site (m-w).** The vertical dashed line indicates zero error. Sample sizes for each site are provided in Table 1.
(TIF)

**S2 Fig. Empirical cumulative distribution function (ECDF) plots comparing observed and model-generated daily wind speed at each calibration (C) site (a-l) and validation (V) site (m-w).** The vertical dashed line indicates zero error. Sample sizes for each site are provided in Table 1.
(TIF)

**S3 Fig. Empirical cumulative distribution function (ECDF) plots comparing the observed and model-generated daily rainfall at each calibration site (a-l) and validation site (m-w).** The vertical dashed line indicates zero error. Sample sizes for each site are provided in Table 1.
(TIF)

**S4 Fig. Validation of wet and dry spell length distributions generated by the first-order Markov chain.** Box plots show distributions of (a) wet spell lengths (n = 6928 observed, 7111 generated) and (b) dry spell lengths (n = 6928 observed, 7125 generated) pooled from all individual spell occurrences across all 23 sites. Box plots display median (center line), interquartile range (IQR) (box), 1.5 × IQR whiskers, and outliers (points). Empirical cumulative distribution functions (ECDF) show (c) wet spell and (d) dry spell distributions. Blue represents observed (obs.) data; red represents generated (gen.) data.
(TIF)

**S5 Fig. Site-level comparison of wet and dry spell length statistics from first-order Markov chain (n = 23 sites).** Each point represents summary statistics calculated separately for one site from both observed (obs.) and generated (gen.) data. Panels show (a) mean wet spell length per site, (b) mean dry spell length per site, (c) maximum wet spell length observed at each site, and (d) maximum dry spell length observed at each site. The dashed line indicates perfect agreement (1:1 line). Blue and red markers represent wet spell and dry spell statistics, respectively. (TIF)

## Author contributions

**Conceptualization:** Christopher Boon Sung Teh.

**Data curation:** Christopher Boon Sung Teh, See Siang Cheah, David Ross Appleton.

**Formal analysis:** Christopher Boon Sung Teh.

**Funding acquisition:** Christopher Boon Sung Teh, David Ross Appleton.

**Investigation:** Christopher Boon Sung Teh.

**Methodology:** Christopher Boon Sung Teh.

**Project administration:** Christopher Boon Sung Teh.

**Resources:** Christopher Boon Sung Teh, See Siang Cheah, David Ross Appleton.

**Software:** Christopher Boon Sung Teh.

**Validation:** Christopher Boon Sung Teh, See Siang Cheah, David Ross Appleton.

**Visualization:** Christopher Boon Sung Teh.

**Writing – original draft:** Christopher Boon Sung Teh.

**Writing – review & editing:** Christopher Boon Sung Teh, See Siang Cheah.

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
