## [Decision Letter · Decision Letter 0]

26 Sep 2025

Dear Dr.  Teh,

We look forward to receiving your revised manuscript.

Kind regards,

Salim Heddam

Academic Editor

PLOS ONE

Journal Requirements:

“Research Incentive Grant for Teaching and Learning (GIPP) 2024 Project code 9323794”

Additional Editor Comments:

Reviewer 1#:

Dear authors,

I have just few comments to your interesting manuscript:

1) Please, check the keywords: L. 39 there should be "distribution", and not use just abbreviation "MsiaGen".

2) In the chapter Study Sites, L. 87, please introduce there why just 23 stations are considered (it is later in L. 102-103.

Please, introduce there the climate zonation according to the Koppen classification, the potential evapotranspiration

and a potential water budget.

3) L. 220, please, introduce the definition of dry and wet days.

4) The model Msia Gen should be well described in Methodology, do not do it in Conclusions (L.767-770 are not necessary),

in the chapter Conclusions go directly to your achievements.

Reviewer 2#:

The manuscript introduces MsiaGen, a stochastic daily weather generator tailored for Malaysia’s tropical climate. It emphasizes computational simplicity, use of daily data (instead of hourly), site-specific calibration, and application in oil palm yield modeling. The model employs Skew Normal distributions for temperature, Weibull for wind speed, and Gamma (with GEV-derived shape) for rainfall, with temporal dependence modeled via AR(1) processes and a first-order Markov chain. Validation across 23 sites shows good agreement with observations, though performance weakens for extremes (heavy rainfall and high wind speeds).

Strengths: The paper addresses the lack of simple, robust weather generators designed for tropical agricultural contexts. The paper also adapts statistical techniques (Skew Normal, Weibull, Gamma+GEV) to Malaysian climate, bridging a gap between advanced but data-hungry generators (e.g., AWE-GEN) and practical agricultural applications. Multi-site calibration/validation, metrics (NMAE, NMBE, KGE, AD test), and diagnostic plots (QQ, ECDF, hex-bin) are used for validation. Coupling MsiaGen with Sawit.jl for oil palm yield simulations strengthens the case for applied agricultural relevance.

Weaknesses: Only 3–5 years of data per site. This limits robustness, particularly for extremes. Authors acknowledge this, but stronger justification (or sensitivity analysis) would help. The Gamma distribution + Markov chain may not adequately capture heavy-tailed rainfall and long wet/dry spells. Authors briefly discuss alternatives (higher-order Markov, non-Gamma models), but don’t attempt even exploratory comparison. While validated across sites, it is unclear how MsiaGen performs outside Malaysia (e.g., equatorial Southeast Asia more broadly). This could affect generalizability. The choice of AR(1) and first-order Markov is justified for parsimony, but at the cost of realism during monsoon persistence. More quantitative discussion of wet/dry spell length reproduction is warranted. Since calibration sites had lower rainfall extremes than validation sites, performance differences may partly stem from this imbalance. A stratified validation approach might yield clearer insights. The manuscript is dense in statistical detail, which may overwhelm readers unfamiliar with distribution fitting. A summary table of all chosen distributions, parameters, and rationales would improve readability. Figures are strong, but some (ECDF panels for each site) are too numerous for the main text—consider moving to supplementary materials.

Suggestions: Test an alternative rainfall distribution (e.g., Generalized Pareto, mixed-exponential) for comparison, even if briefly, to show MsiaGen’s trade-offs. Add explicit metrics for wet/dry spell length reproduction and compare to observations. If possible, include one or two sites with ≥20 years of data to test parameter stability. Provide more detail on how inaccuracies in extremes (e.g., rainfall >100 mm) might impact oil palm yield modeling outcomes. Move highly technical derivations (e.g., parameter transformations for Skew Normal) to appendices/supplementary material. This keeps the main text accessible. Typo in keywords (“distibution” → “distribution”), improve figure captions (add sample sizes, error bars). The study is well-motivated and promising, but before acceptance the authors should:

• Better address the limitations in modeling extremes and persistence,

• Provide clearer evidence of MsiaGen’s stability and generalizability, and

• Streamline statistical detail for readability.

Comments: The calibration is based on relatively short weather records (3–5 years per site). This limitation likely contributes to underperformance in simulating extremes. It may be justified more clearly why this truncation was necessary and, if possible, provide a sensitivity analysis or at least include one or two sites with longer datasets to test parameter stability. The manuscript acknowledges underestimation of high rainfall (>100 mm) and wind speeds (>3–4 m s⁻¹). Since these extremes are critical for agriculture, especially for perennial crops like oil palm, the authors should:

• Provide explicit quantitative evaluation of how frequently such events are misrepresented.

• Explore or at least compare with alternative distributions (e.g., Generalized Pareto, mixed-exponential) that may better capture heavy tails.

The first-order Markov chain may inadequately represent monsoonal persistence. The discussion notes this but does not present quantitative validation. Spell length statistics (mean wet/dry spell length, frequency distribution) comparing observed vs. simulated data may be included. This would greatly strengthen confidence in MsiaGen’s ability to reproduce tropical rainfall characteristics. While the model is designed for Malaysia, it would be valuable to discuss explicitly whether MsiaGen can be extended to other equatorial regions with similar Af climates (e.g., Indonesia, Papua New Guinea). Even if not tested, a short discussion would help readers assess applicability. The yield simulations show close agreement overall, but the manuscript does not quantify how misrepresentation of extremes propagates into yield estimates. The following would need clarification:

• Do errors in extreme rainfall or wind events materially affect modelled yields?

• Could yield risk assessments underestimate extreme-event-driven yield loss?

The level of mathematical detail (e.g., parameter derivations for the Skew Normal distribution) may overwhelm readers. Consider moving some of these derivations to supplementary material and instead provide a concise summary table of distributions, parameters, and rationales in the main text. Figures showing ECDFs for every site are informative but numerous. Suggest moving some panels to Supplementary Information and retaining representative sites in the main text. Add sample sizes and error bars (where applicable) in figure captions. The manuscript states that full datasets are available but relies heavily on short station records. It may be clarified whether raw station data are archived in a national database and if readers can access them directly, or whether only processed datasets are provided.

Recommendation: The manuscript makes a valuable contribution to agricultural modeling in tropical environments. However, I recommend major revisions to address the limitations in extreme event representation, spell length validation, and presentation clarity. Once revised, the paper will be a strong candidate for publication.

Reviewer's Responses to Questions

**Comments to the Author**

1. Is the manuscript technically sound, and do the data support the conclusions?

Reviewer #1: Yes

Reviewer #2: Partly

2. Has the statistical analysis been performed appropriately and rigorously?

Reviewer #1: Yes

Reviewer #2: Yes

3. Have the authors made all data underlying the findings in their manuscript fully available?

Reviewer #1: Yes

Reviewer #2: Yes

4. Is the manuscript presented in an intelligible fashion and written in standard English?

Reviewer #1: Yes

Reviewer #2: Yes

Reviewer #1: Dear authors,

I have just few comments to your interesting manuscript:

1) Please, check the keywords: L. 39 there should be "distribution", and not use just abbreviation "MsiaGen".

2) In the chapter Study Sites, L. 87, please introduce there why just 23 stations are considered (it is later in L. 102-103.

Please, introduce there the climate zonation according to the Koppen classification, the potential evapotranspiration

and a potential water budget.

3) L. 220, please, introduce the definition of dry and wet days.

4) The model Msia Gen should be well described in Methodology, do not do it in Conclusions (L.767-770 are not necessary),

in the chapter Conclusions go directly to your achievements.

Reviewer #2: The manuscript introduces MsiaGen, a stochastic daily weather generator tailored for Malaysia’s tropical climate. It emphasizes computational simplicity, use of daily data (instead of hourly), site-specific calibration, and application in oil palm yield modeling. The model employs Skew Normal distributions for temperature, Weibull for wind speed, and Gamma (with GEV-derived shape) for rainfall, with temporal dependence modeled via AR(1) processes and a first-order Markov chain. Validation across 23 sites shows good agreement with observations, though performance weakens for extremes (heavy rainfall and high wind speeds).

Strengths: The paper addresses the lack of simple, robust weather generators designed for tropical agricultural contexts. The paper also adapts statistical techniques (Skew Normal, Weibull, Gamma+GEV) to Malaysian climate, bridging a gap between advanced but data-hungry generators (e.g., AWE-GEN) and practical agricultural applications. Multi-site calibration/validation, metrics (NMAE, NMBE, KGE, AD test), and diagnostic plots (QQ, ECDF, hex-bin) are used for validation. Coupling MsiaGen with Sawit.jl for oil palm yield simulations strengthens the case for applied agricultural relevance.

Weaknesses: Only 3–5 years of data per site. This limits robustness, particularly for extremes. Authors acknowledge this, but stronger justification (or sensitivity analysis) would help. The Gamma distribution + Markov chain may not adequately capture heavy-tailed rainfall and long wet/dry spells. Authors briefly discuss alternatives (higher-order Markov, non-Gamma models), but don’t attempt even exploratory comparison. While validated across sites, it is unclear how MsiaGen performs outside Malaysia (e.g., equatorial Southeast Asia more broadly). This could affect generalizability. The choice of AR(1) and first-order Markov is justified for parsimony, but at the cost of realism during monsoon persistence. More quantitative discussion of wet/dry spell length reproduction is warranted. Since calibration sites had lower rainfall extremes than validation sites, performance differences may partly stem from this imbalance. A stratified validation approach might yield clearer insights. The manuscript is dense in statistical detail, which may overwhelm readers unfamiliar with distribution fitting. A summary table of all chosen distributions, parameters, and rationales would improve readability. Figures are strong, but some (ECDF panels for each site) are too numerous for the main text—consider moving to supplementary materials.

Suggestions: Test an alternative rainfall distribution (e.g., Generalized Pareto, mixed-exponential) for comparison, even if briefly, to show MsiaGen’s trade-offs. Add explicit metrics for wet/dry spell length reproduction and compare to observations. If possible, include one or two sites with ≥20 years of data to test parameter stability. Provide more detail on how inaccuracies in extremes (e.g., rainfall >100 mm) might impact oil palm yield modeling outcomes. Move highly technical derivations (e.g., parameter transformations for Skew Normal) to appendices/supplementary material. This keeps the main text accessible. Typo in keywords (“distibution” → “distribution”), improve figure captions (add sample sizes, error bars). The study is well-motivated and promising, but before acceptance the authors should:

• Better address the limitations in modeling extremes and persistence,

• Provide clearer evidence of MsiaGen’s stability and generalizability, and

• Streamline statistical detail for readability.

Comments: The calibration is based on relatively short weather records (3–5 years per site). This limitation likely contributes to underperformance in simulating extremes. It may be justified more clearly why this truncation was necessary and, if possible, provide a sensitivity analysis or at least include one or two sites with longer datasets to test parameter stability. The manuscript acknowledges underestimation of high rainfall (>100 mm) and wind speeds (>3–4 m s⁻¹). Since these extremes are critical for agriculture, especially for perennial crops like oil palm, the authors should:

• Provide explicit quantitative evaluation of how frequently such events are misrepresented.

• Explore or at least compare with alternative distributions (e.g., Generalized Pareto, mixed-exponential) that may better capture heavy tails.

The first-order Markov chain may inadequately represent monsoonal persistence. The discussion notes this but does not present quantitative validation. Spell length statistics (mean wet/dry spell length, frequency distribution) comparing observed vs. simulated data may be included. This would greatly strengthen confidence in MsiaGen’s ability to reproduce tropical rainfall characteristics. While the model is designed for Malaysia, it would be valuable to discuss explicitly whether MsiaGen can be extended to other equatorial regions with similar Af climates (e.g., Indonesia, Papua New Guinea). Even if not tested, a short discussion would help readers assess applicability. The yield simulations show close agreement overall, but the manuscript does not quantify how misrepresentation of extremes propagates into yield estimates. The following would need clarification:

• Do errors in extreme rainfall or wind events materially affect modelled yields?

• Could yield risk assessments underestimate extreme-event-driven yield loss?

The level of mathematical detail (e.g., parameter derivations for the Skew Normal distribution) may overwhelm readers. Consider moving some of these derivations to supplementary material and instead provide a concise summary table of distributions, parameters, and rationales in the main text. Figures showing ECDFs for every site are informative but numerous. Suggest moving some panels to Supplementary Information and retaining representative sites in the main text. Add sample sizes and error bars (where applicable) in figure captions. The manuscript states that full datasets are available but relies heavily on short station records. It may be clarified whether raw station data are archived in a national database and if readers can access them directly, or whether only processed datasets are provided.

Recommendation: The manuscript makes a valuable contribution to agricultural modeling in tropical environments. However, I recommend major revisions to address the limitations in extreme event representation, spell length validation, and presentation clarity. Once revised, the paper will be a strong candidate for publication.

**Do you want your identity to be public for this peer review?** For information about this choice, including consent withdrawal, please see our Privacy Policy

Reviewer #1: No

Reviewer #2: **Yes:**  Muttath Leo Franklin

---

## [Author Response · Author response to Decision Letter 1]

14 Oct 2025

We apologize for the confusion regarding the mismatch between the "Funding Information" and "Financial Disclosure" sections mentioned in your letter.

During the revision submission process, we were unable to locate the "Financial Disclosure" field in the submission system. As a result, we could not update this section to match our funding information.

For clarity, here is our complete funding information:

Our study was funded by Universiti Putra Malaysia under the "Research Incentive Grant for Teaching and Learning (GIPP) 2024," Project code no. 9323794. The funders had no role in study design, data collection and analysis, decision to publish, or preparation of the manuscript.

We have included this funding statement in our cover letter. If you could direct us to the specific location of the "Financial Disclosure" field in the submission system, we would be happy to update it accordingly to ensure consistency across all sections.

---

## [Decision Letter · Decision Letter 1]

2 Nov 2025

Dear Dr. Teh,

We look forward to receiving your revised manuscript.

Kind regards,

Salim Heddam

Academic Editor

PLOS ONE

**Journal Requirements:**

**Additional Editor Comments:**

**Reviewer 1#** :Dear authors,

Thank you for submitting your interesting manuscript. I have just the following comments:

1) Please, keep the limit of 300 words in the abstract.

2) In keywords, I think, it is better to use "oil palm yield".

3) Considering the structure of your paper, probably, better seems to use the standard structure required by the journal

guidlines: Material and methods ....

4) I prefer the chapter "Study site" (instead Study sites), descripting the area of your interest, those 23 meteorological stations

are just parts of this study site.

5) Please discuss the honogeinity of the analysed data-sets.

6) And, discuss also when the model performance has comparable value in all the climate district of your study.

7) The statement "MsiaGen is written in Julia programming language, and its full source code is available at:

github.com/cbsteh/MsiaGen" belongs not to the Conclusions, it should be introduced in the chapter describing the model.

**Reviewer 2#:** The development of MsiaGen fills a critical gap for tropical agricultural applications, where daily data are scarce and models like AWE-GEN are impractical.

The use of Skew Normal distributions for temperature and GEV-based variability in rainfall shape parameters is innovative and well justified.

The model was tested across 23 diverse sites, and validation metrics (KGE > 0.8; NMAE < 2.5%) are impressive.

The dual-scale evaluation (monthly and daily) and multiple diagnostics (KGE, QQ, ECDF, AD tests) demonstrate thorough statistical rigor.

The independent test via Sawit.jl simulations adds strong applied validation.

The manuscript is well structured, with equations clearly presented.

Data and model code availability via GitHub enhances transparency.

The manuscript is recommended for publication

Reviewers' comments:

Reviewer's Responses to Questions

**Comments to the Author**

Reviewer #1: (No Response)

Reviewer #2: All comments have been addressed

2. Is the manuscript technically sound, and do the data support the conclusions?

Reviewer #1: Yes

Reviewer #2: Yes

3. Has the statistical analysis been performed appropriately and rigorously?

Reviewer #1: Yes

Reviewer #2: N/A

4. Have the authors made all data underlying the findings in their manuscript fully available?

Reviewer #1: Yes

Reviewer #2: Yes

5. Is the manuscript presented in an intelligible fashion and written in standard English?

Reviewer #1: Yes

Reviewer #2: Yes

**Reviewer #1:**  Dear authors,

Thank you for submitting your interesting manuscript. I have just the following comments:

1) Please, keep the limit of 300 words in the abstract.

2) In keywords, I think, it is better to use "oil palm yield".

3) Considering the structure of your paper, probably, better seems to use the standard structure required by the journal

guidlines: Material and methods ....

4) I prefer the chapter "Study site" (instead Study sites), descripting the area of your interest, those 23 meteorological stations

are just parts of this study site.

5) Please discuss the honogeinity of the analysed data-sets.

6) And, discuss also when the model performance has comparable value in all the climate district of your study.

7) The statement "MsiaGen is written in Julia programming language, and its full source code is available at:

github.com/cbsteh/MsiaGen" belongs not to the Conclusions, it should be introduced in the chapter describing the model.

**Reviewer #2:** The development of MsiaGen fills a critical gap for tropical agricultural applications, where daily data are scarce and models like AWE-GEN are impractical.

The use of Skew Normal distributions for temperature and GEV-based variability in rainfall shape parameters is innovative and well justified.

The model was tested across 23 diverse sites, and validation metrics (KGE > 0.8; NMAE < 2.5%) are impressive.

The dual-scale evaluation (monthly and daily) and multiple diagnostics (KGE, QQ, ECDF, AD tests) demonstrate thorough statistical rigor.

The independent test via Sawit.jl simulations adds strong applied validation.

The manuscript is well structured, with equations clearly presented.

Data and model code availability via GitHub enhances transparency.

The manuscript is recommended for publication

**Do you want your identity to be public for this peer review?** For information about this choice, including consent withdrawal, please see our Privacy Policy

Reviewer #1: No

Reviewer #2: No

---

## [Author Response · Author response to Decision Letter 2]

7 Nov 2025

We have made corrections/amendments to every point raised by Reviewer #1 (note: Reviewer #2 did not have further comments). Please see "Response 2 to the reviewers.docx" file for our amendments.

---

## [Decision Letter · Decision Letter 2]

5 Jan 2026

A stochastic daily weather generator for perennial crop simulations in tropical Malaysia

PONE-D-25-41380R2

Dear Dr. Teh,

We’re pleased to inform you that your manuscript has been judged scientifically suitable for publication and will be formally accepted for publication once it meets all outstanding technical requirements.

Kind regards,

Lingye Yao, Ph.D.

Academic Editor

PLOS One

Additional Editor Comments (optional):

Reviewers' comments:

Reviewer's Responses to Questions

**Comments to the Author**

Reviewer #1: All comments have been addressed

Reviewer #3: All comments have been addressed

2. Is the manuscript technically sound, and do the data support the conclusions?

Reviewer #1: Yes

Reviewer #3: Yes

3. Has the statistical analysis been performed appropriately and rigorously?

Reviewer #1: Yes

Reviewer #3: Yes

4. Have the authors made all data underlying the findings in their manuscript fully available?

Reviewer #1: Yes

Reviewer #3: Yes

5. Is the manuscript presented in an intelligible fashion and written in standard English?

Reviewer #1: Yes

Reviewer #3: Yes

Reviewer #1: Dear authors,

thank you for your kind revision of your paper, I very much appreciate your personal effort to improve the quality of your paper..

Reviewer #3: Thank you for submitting the revised manuscript and for the diligent effort you have made in addressing the reviewer's comments. I appreciate the time and effort you have devoted to revising the manuscript.

**Do you want your identity to be public for this peer review?** For information about this choice, including consent withdrawal, please see our Privacy Policy

Reviewer #1: No

Reviewer #3: No

---

## [Editor Report · Acceptance letter]

PONE-D-25-41380R2

PLOS One

Dear Dr. Teh,

I'm pleased to inform you that your manuscript has been deemed suitable for publication in PLOS One. Congratulations! Your manuscript is now being handed over to our production team.

Kind regards,

on behalf of

Dr. Lingye Yao

Academic Editor

PLOS One